materials science

hybrid, organic–metallic, synergistic effect, intumescent flame retardant, ethylene-vinyl acetate

**Authors for correspondence:**
Bo Xu
e-mail: xubo@btbu.edu.cn
Lijun Qian
e-mail: qianlj@btbu.edu.cn

This article has been edited by the Royal Society of Chemistry, including the commissioning, peer review process and editorial aspects up to the point of acceptance.

# Enhancement of an organic–metallic hybrid charring agent on flame retardancy of ethylene-vinyl acetate copolymer

Bo Xu[1,2], Wen Ma[1,2], Lushan Shao[1,2], Lijun Qian[1,2] and Yong Qiu[1,2]

[1]School of Materials Science and Mechanical Engineering, and [2]Beijing Key Laboratory of Quality Evaluation Technology for Hygiene and Safety of Plastics, Beijing Technology and Business University, Fucheng Road 11, Beijing 100048, People's Republic of China

BX, 0000-0001-9114-4015

An organic triazine charring agent hybrid with zinc oxide (OTCA@ZnO) was prepared and well characterized through Fourier transform infrared spectrometry (FTIR), solid-state nuclear magnetic resonance (SSNMR), transmission electron microscopy (TEM) and thermogravimetric analysis (TGA). The flame retardancy and thermal behaviour of intumescent flame retardant ethylene-vinyl acetate (EVA) composites combining OTCA@ZnO and ammonium polyphosphate (APP) were investigated using limited oxygen index (LOI), UL-94 vertical burning, cone calorimetry and TGA. The structure and morphology of chars were investigated by scanning electron microscopy (SEM), FTIR, laser Raman spectroscopy analysis (LRS) and X-ray photoelectron spectroscopy (XPS). Results revealed that OTCA@ZnO exhibited excellent thermal stability and dispersity after hybridization. The flame retardancy and smoke suppression properties of EVA were significantly improved by introducing APP/OTCA@ZnO. TGA results indicated that APP/OTCA@ZnO presented an excellent synergistic effect and promoted the char formation of EVA composites. Residue analysis results showed more char with high quality connected by richer P–O–C, P–N and P–O–Si structures was formed in APP/OTCA@ZnO system than APP/HOTCA/ZnO system, which consequently suppressed more efficiently the combustion and smoke production due to the *in situ* catalytic carbonization effect of hybrid.

# 1. Introduction

Ethylene-vinyl acetate copolymer (EVA) as a copolymer of ethylene and vinyl acetate has been extensively applied in lots of fields, such as gas pipe, insulating materials and so on, owing to its excellent easy-processing, physical and mechanical properties. However, the same as the typical organic polymers, EVA is also inherently flammable with a high smoke emission which restricted its development and application. The halogen-containing flame retardants were excellent flame retardant additives for EVA, especially used with $Sb_2O_3$, and yet corrosive smoke and toxic gases would be frequently generated during combustion which unavoidably caused great harm to the humans' health and environment [1,2]. Therefore, a growing demand for halogen-free and environment-friendly flame retardants (FRs) gradually rises up. Some hydroxides and nanometre materials are widely used for flame retardant EVA due to their excellent low smoke and non-toxic properties. However, excessive loading can cause a decrease in the mechanical properties of the material and in flame retardant efficiency. Therefore, many researchers have recently improved the flame retardancy of polymers by modifying flame retardants or combining with other agents. Vahabia *et al.* [3] applied hydroxyapatite (HA) and a physical blend of HA and multi-walled carbon nanotube (HA-CNT) into EVA, and found the peak heat release rate (PHRR) value of ethylene-vinyl acetate/ammonium polyphosphate/ hydroxyapatite (EVA/APP/HA) or EVA/APP/HA-CNTs was decreased dramatically compared with pure EVA. Kim *et al.* [4] investigated the synergetic effect of a clay-organic intumescent hybrid system of EVA nanocomposites and the flame retardant property of EVA was enhanced. Qian *et al.* [5] prepared the ethylene-vinyl acetate/layered double hydroxides/zinc borate (EVA/LDHs/ZB) composite, and found the ternary composites possessed a higher thermal stability than the EVA/LDH composites.

Among these halogen-free flame retardants, intumescent flame retardants (IFRs) with the advantages of non-toxicity, environmental and low smoke have become one of the best candidates in flame retardant polymer fields in recent years [6,7]. The existing results showed that the special IFRs with triazine charring agents, which exhibited high flame retardancy because of the stable triazine ring structure and easily charring characteristics during combustion, have been extensively investigated and they could overcome the shortcomings of the traditional IFRs [8–10].

Further, for enhancing the efficiency of intumescent systems on polymers, lots of additive agents, such as metallic oxides, 4A zeolite, fumed silica and so on, have been widely studied. It was reported that a small amount of metal oxides (e.g. ZnO and $La_2O_3$) could significantly improve the thermal stability and flame retardancy of intumescent flame retardant composites [11–13]. During combustion, metallic oxides played roles of catalysing the decomposition and charring reaction of IFRs, and more stable cross-linked structures formed in intumescent char layer [14,15]. Lewin *et al.* [16] studied the catalytic effect of several metallic compounds in IFR systems. Sheng *et al.* [17] investigated the com-ZnO, nano-ZnO, zeolite 4A and $Al(H_2PO_2)_3$ as synergistic agents combined with IFR in PP, and Wu & Yang [18] investigated some transition metal oxides like $MnO_2$, ZnO and $Ni_2O_3$, which were applied into PP/IFR system. The results indicated it was the interaction between APP and metal oxides that enhanced the flame retardant properties of polymer. However, the existing works showed metal oxides were usually blended physically with IFR in polymers, and thus it was difficult for the mixtures to fully show their synergism due to the very low loading amounts (less than 2 wt%) and serious reunion of oxides. The final properties of polymeric composites strongly depend on particles' dispersion and distribution in the matrix [19]. Moreover, the weak compatibility of metal oxides in composites also was a severe disadvantage, which deteriorated the physical and mechanical properties [20]. So, the organic surface modification of metal oxides, especially employing those modifiers with fire resistance, may be a prospective method, which would not only settle the above matters, but also bring more efficient synergistic effect of metal oxides within one compound unit with flame retardants.

Silicon-containing compounds are types of environment-friendly and highly efficient flame retardants which could form a protective carbonaceous layer with excellent barrier effect at high temperature [21–24]. It was reported that silicone could comparatively lower the heat release rate and release product of toxic gas (CO), and the synergistic effect was especially obvious between silicone and IFR with the protective char layer containing Si–O and Si–C bonds [23,24]. Further, silicones could improve the interfacial compatibility of fillers and matrix by chemical interactions, which regulated the structure and property of composites [25,26].

In this work, based on the flame retardant effectiveness of triazine compounds and silicon, and the synergistic effect of ZnO, a hybrid organic–metallic charring agent (OTCA@ZnO) had been prepared. The flame retardancy and thermal stability of EVA composites containing APP and OTCA@ZnO were studied through LOI, UL-94, cone test and TGA. The residues were studied to explore the flame

**Scheme 1.** Route for the synthesis of OTCA.

retardant mechanism by SEM, FTIR, LRS and XPS. APP/HOTCA/ZnO system as a comparing subject of APP/OTCA@ZnO system was also systematically investigated to identify the more efficient flame retardant action of the hybrid system.

# 2. Experimental procedure

## 2.1. Materials

EVA containing vinyl acetate units of 17.6–20.4 wt% was bought from BASF-YPC Co., Ltd (Nanjing, China). 2,4,6-Trichoro-1,3,5-triazine (CYC) and ethylenediamine (EDA) were supplied by Tianjin Hebei Chengxin Chemical Co. Ltd and Fuchen Chemical Reagents Factory, China, respectively, and used as received. APP ($n > 1000$) was produced by Polyrocks Chemical Co. Ltd, China. γ-Aminopropyltriethoxy silane, N,N-dimethyl acetamide (DMAc), dichloromethane (DCM), triethylamine (TEA) and nano-ZnO were provided by Sinopharm Chemical Reagent Co., Ltd, China. DMAc was used after purification.

## 2.2. Preparation of OTCA@ZnO

Firstly, 20 ml DMAc solution containing 32 g γ-aminopropyltriethoxy silane was added into a three-necked flask equipped with a mechanical stirrer. Then, 27 g CYC and 14.6 g TEA in 20 ml DMAc was slowly added dropwise with vigorously stirring in an external ice-water bath. The reaction temperature was increased to 50°C in an oil bath after 3 h and a solution of 19.2 g EDA and 14.6 g TEA in 10 ml DMAc was dropped for 30 min. After 3 h, 14.6 g TEA was added and the reaction temperature increased to 95–100°C for another 3 h. Thereafter, the mixture after vacuum distillation was cooled and poured into 50 ml DCM. The precipitate was then filtered and washed with DCM. After drying to a constant weight under vacuum at 100°C, the intermediate OTCA (light yellow powder) was obtained. Scheme 1 presents the routes of synthesis for OTCA. FTIR (KBr, cm$^{-1}$): 3012 (CH$_3$), 2925 (CH$_2$), 1586 and 1512 (C$_3$N$_3$), 1028–1005 (Si–O), 813 (Si–C).

Secondly, 200 ml mixed solution of ethanol/water (v/v 9/1) and 32 g OTCA, with pH altered to 9 with 5 wt% ammonia solution, was stirred for 2 h. Meanwhile, 8 g nano-ZnO activated at 150°C for 2 h and 150 ml ethanol were added into a conical flask and ultrasonic agitation was carried out for 2 h. Then, the mixture was dropped into OTCA/ethanol/water solution and refluxed gently with stirring for 6 h. Next, the reaction mixture was cooled to room temperature and filtered to obtain raw product. The solid was washed three times with hot water. Finally, after drying at 100°C under vacuum overnight, a white powder was obtained (yield: 86%). Scheme 2 presents the preparation route of OTCA@ZnO. The content of Zn in OTCA@ZnO was 7.2 wt% tested by ICP-MS (iCAP Q, Thermo, Waltham, USA) and the content of ZnO was 9.0 wt% calculated by Zn content. FTIR (KBr, cm$^{-1}$): 3427 (OH), 2929 (CH$_2$), 1586 and 1513 (C$_3$N$_3$), 1083 (Si–O), 809 (Si–C), 434 (Si–O–ZnO). $^{13}$C SSNMR (Solid, ppm): $\delta = 168$ and 76 (s, 3C, C$_3$N$_3$), $\delta = 44$ (s, 6C, NH–CH$_2$), $\delta = 26$ (s, 2C, CH$_2$), $\delta = 14$ (s, 2C, Si–CH$_2$). $^{29}$Si SSNMR (Solid, ppm): $\delta = -54$ (m, 1Si, Si–O–ZnO), $\delta = -57$ (m, 2Si, Si–O–Si). The corresponding spectra of OTCA and OTCA@ZnO were given in the electronic supplementary material.

Meanwhile, the hydrolysis product of OTCA named as HOTCA was prepared following the above procedure except without ZnO, and HOTCA physically blended with nano-ZnO would be used as a comparative object of the hybrid OTCA@ZnO.

## 2.3. Preparation of flame retardant EVA composites

EVA and all additives were dried in vacuum at 80°C overnight. EVA samples were melt-mixed via a torque rheometer (KX-160, Jiangsu, China) with the process temperature of 130°C and the rotation

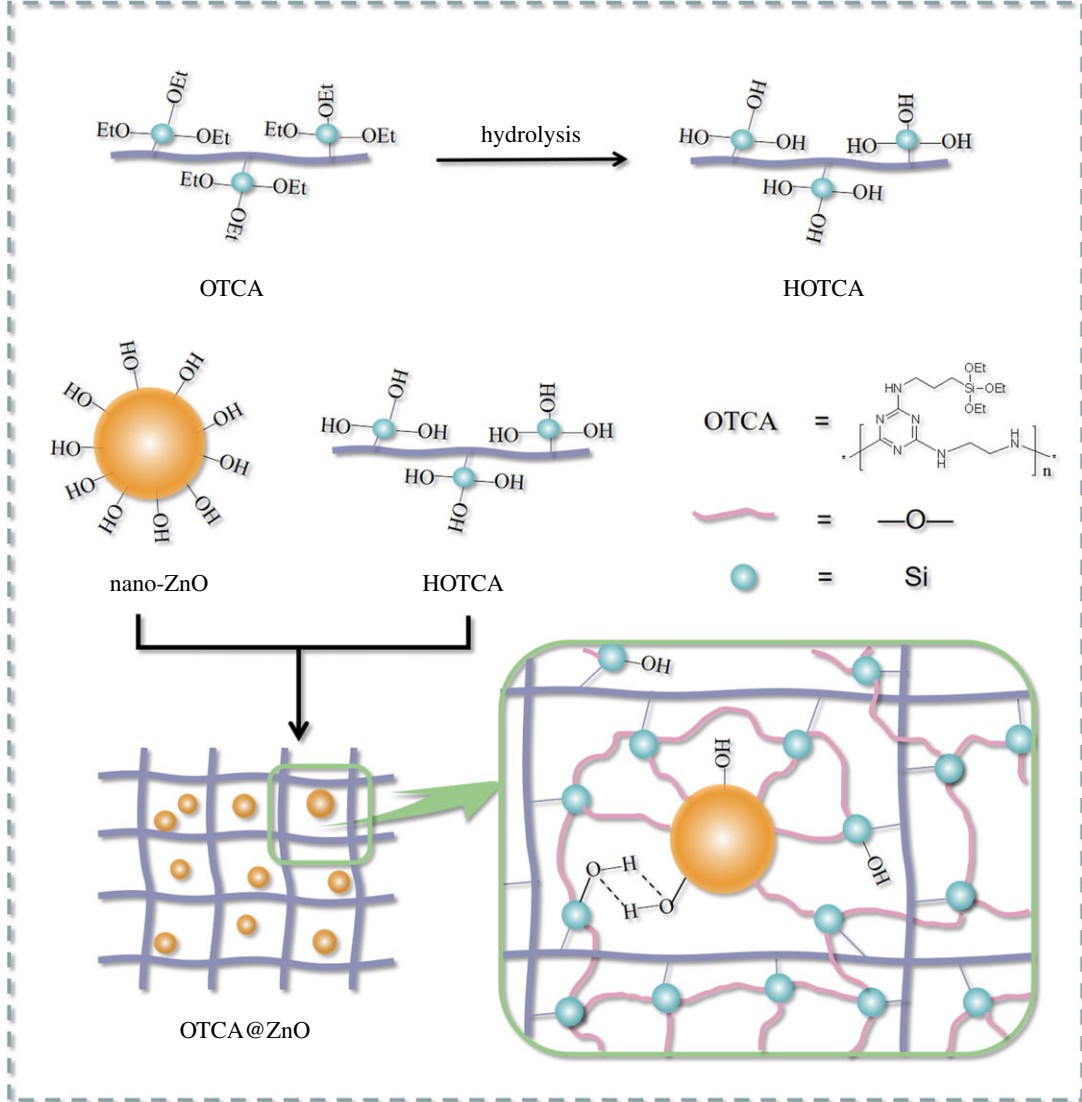

**Scheme 2.** Route for the synthesis of OTCA@ZnO.

speed of 40 r.p.m. Then, they were hot-pressed to produce sheets with different sizes used in subsequent tests. The formulations are presented in table 1.

## 2.4. Instruments and characterization

Fourier transform infrared (FTIR) spectra of compounds and residues after cone test were recorded by a Nicolet 6700 FTIR spectrometer (Nicolet, USA) with KBr pellets. $^1$H NMR, $^{13}$C NMR and $^{29}$Si NMR spectra were obtained from an Agilent 600 M spectrometer (Agilent, America). Transmission electron microscopy (TEM) images were taken on JEM-2100 (JEOL, Japan) with 200 kV acceleration.

Thermogravimetric analysis (TGA) tests were performed on a TGA instrument thermal analyser (TA Q50, USA). About 2–5 mg of a tested sample was heated from 50°C to 700°C under neat $N_2$ atmosphere at rate of 20°C min$^{-1}$.

Limited oxygen index (LOI) values were measured using a FTT 0080 LOI instrument (Fire Testing Technology Ltd (FTT), UK) according to American Society for Testing and Materials (ASTM) D2863-17 and the sample dimension was $130 \times 6.5 \times 3.2$ mm. UL-94 vertical burning rating tests were performed by an FTT0082 tester (FTT, UK) following the procedures in ASTM D3801−10 (sample dimension: $125 \times 12.7 \times 3.2$ mm). The forced combustion behaviour was measured by an FTT0007 cone calorimetry (FTT, UK) according to ISO 5660-1 with an external heat flux of 50 kW m$^{-2}$ (sample dimension: $100 \times 100 \times 3$ mm).

**Table 1.** Formulation and flammability analysis (LOI and UL-94) of EVA and EVA composites.

| samples | components (wt%) | | | | | LOI (%) | UL-94 | | |
|---|---|---|---|---|---|---|---|---|---|
| | EVA | APP | OTCA@ZnO | HOTCA | ZnO | | $t_1$ (s)/$t_2$ (s)[a] | dripping | ranking |
| EVA | 100.00 | 0.00 | 0.00 | 0.00 | 0.00 | 17.0 | burn[b] | yes/yes[c] | NR |
| EVA/APP | 75.00 | 25.00 | 0.00 | 0.00 | 0.00 | 25.8 | 10.9/burn | yes/yes | NR |
| EVA/HOTCA | 75.00 | 0.00 | 0.00 | 25.00 | 0.00 | 19.2 | burn | yes/yes | NR |
| EVA/OTCA@ZnO | 75.00 | 0.00 | 25.00 | 0.00 | 0.00 | 19.7 | burn | yes/yes | NR |
| EVA/IFR-1 | 75.00 | 20.00 | 5.00 | 0.00 | 0.00 | 27.6 | 3.2/5.8 | no/no | V-0 |
| EVA/IFR-2 | 75.00 | 18.75 | 6.25 | — | — | 28.8 | 1.4/6.3 | no/no | V-0 |
| EVA/IFR-3 | 75.00 | 16.67 | 8.33 | — | — | 29.7 | 0.0/3.3 | no/no | V-0 |
| EVA/IFR-4 | 75.00 | 12.50 | 12.50 | — | — | 24.3 | burn | yes/yes | NR |
| EVA/IFR-5 | 75.00 | 8.30 | 16.70 | — | — | 24.0 | burn | yes/yes | NR |
| EVA/IFR-6 | 75.00 | 16.67 | — | 7.58 | 0.75 | 27.4 | 6.4/13.5 | no/yes | V-2 |
| EVA/IFR-7 | 75.00 | 16.67 | — | 8.33 | | 25.9 | 10.1/19.9 | yes/yes | V-2 |
| EVA/IFR-8 | 80.00 | 13.33 | 6.67 | — | — | 27.3 | 11.7/21.3 | no/yes | V-2 |

[a]$t_1$ (s)/$t_2$ (s): average time of combustion of five specimens after the first and second ignition.

[b]burn: burning to the clamp.

[c]yes/yes: corresponds to the first/second ignition.

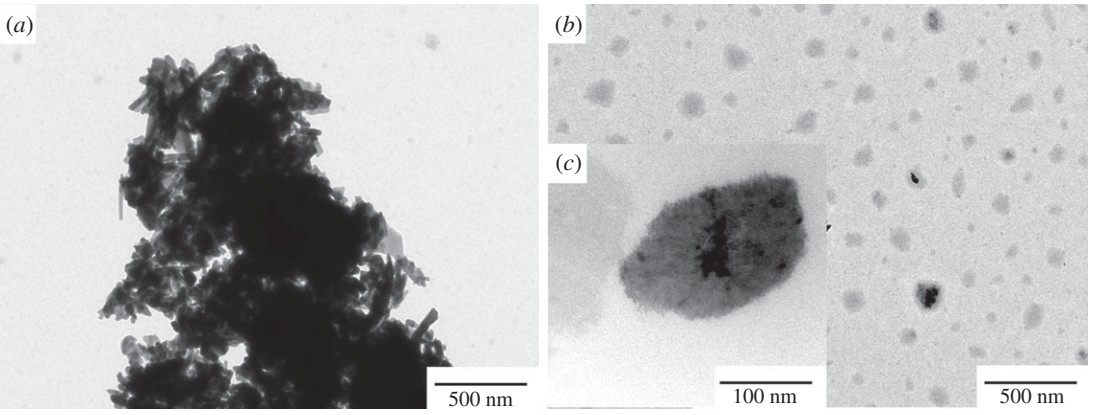

**Figure 1.** TEM images of nano-ZnO (*a*) and OTCA@ZnO (*b,c*).

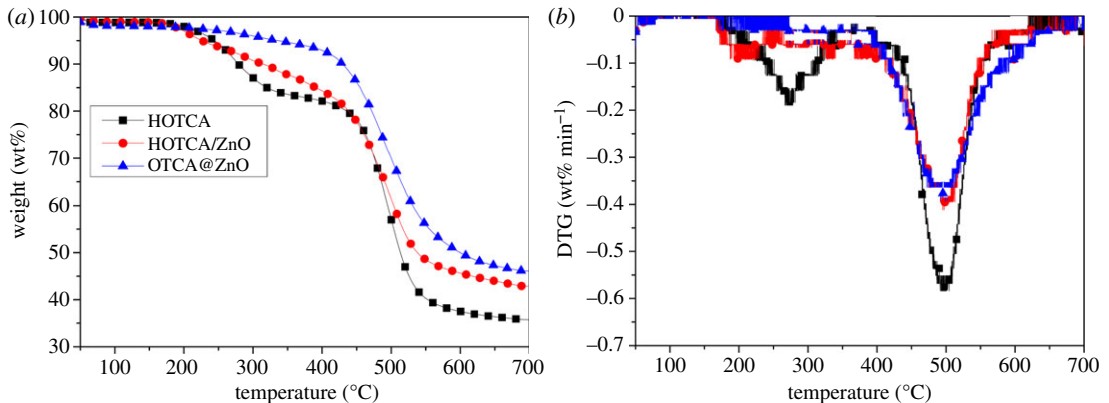

**Figure 2.** TGA (*a*) and DTG (*b*) of HOTCA, HOTCA/ZnO and OTCA@ZnO under $N_2$ atmosphere.

The morphologies of the char were observed in a FEI Quanta 250 FEG field-emission scanning electron microscopy (SEM) with a voltage of 20 kV. Raman spectroscopy analysis (LRS) spectra were obtained from a LabRAM HR Evolution Raman spectrometer (HORIBA, Japan) with excitation by a 532 nm helium-neon laser line and scanning in the range of $50–4000\ cm^{-1}$. X-ray photoelectron spectroscopy (XPS) was recorded by a PHI Quantera-II SXM (Ulvac-PHI, Japan) using Al K$\alpha$ excitation radiation (X-ray power of 2.5 kW).

# 3. Results and discussion

## 3.1. Morphology and thermal properties analysis of OTCA@ZnO

Figure 1 presents TEM images of nano-ZnO and OTCA@ZnO. nano-ZnO possessed shapes like long rods and strongly aggregated into clusters of several micrometres in size due to the high surface energy and lots of hydroxyl groups on the surface. However, the hybrid material presented microstructure-like spheres and the dispersion was greatly improved into nanometre scale. This may be explained that the hybridization consumed the surface hydroxyl groups through the condensation reactions and lowered the surface energy of nano-ZnO. The organic modifier layer covering on surface of nano-ZnO generated the large steric hindrance, obstructing the aggregation of particles. From figure 1, the average particle size of OTCA@ZnO was about 100 nm, which increased from the average particle size of $80 \times 15$ nm of nano-ZnO. Further, most of particles of OTCA@ZnO showed darker centre and lighter edge, which was typical of hybridization structure of inorganic nanometer particles. TEM images also revealed that the well dispersed hybrid containing OTCA and nano-ZnO was successfully prepared.

Figure 2 shows TGA and DTG curves of HOTCA, HOTCA/ZnO and OTCA@ZnO under $N_2$ atmosphere and the corresponding data are listed in table 2. HOTCA had two decomposition steps and the main process took place between 400°C and 550°C. The initial decomposition based on 5% weight loss ($T_{5\%}$) began from 243°C and the char residue was 35.6 wt% at 700°C. The physical blending of nano-

**Table 2.** Thermal properties of HOTCA, HOTCA/ZnO and OTCA@ZnO.

| samples | $T_{5\%}$ (°C) | $T_{50\%}$ (°C) | $T_{peak}$ (°C) | $R_{peak}$ (%/min) | char residue (%) | | |
|---------|-----|-----|-----|-----|-----|-----|-----|
| | | | | | 500°C | 600°C | 700°C |
| HOTCA | 243 | 513 | 500 | 0.59 | 57.1 | 37.5 | 35.6 |
| HOTCA/ZnO | 226 | 538 | 501 | 0.39 | 61.4 | 45.7 | 42.8 |
| OTCA@ZnO | 336 | 599 | 499 | 0.35 | 69.9 | 49.9 | 46.1 |
| APP | 319 | 591 | 565 | 0.55 | 80.1 | 47.3 | 34.2 |
| IFR-3 | 310 | 589 | 499 | 0.32 | 65.0 | 49.3 | 46.3 |
| IFR-3-Cal | 323 | 603 | 586 | 0.41 | 77.2 | 50.9 | 38.2 |

ZnO lowered the $T_{5\%}$ from 243°C for HOTCA to 226°C because of the catalysing effect of the nano-oxide. Yet the char for HOTCA/ZnO was slightly increased (about 1.4 wt%) according to the experimental and calculation values of HOTCA and nano-zinc oxide. As for OTCA@ZnO, the hybridization greatly affected the decomposition behaviour of HOTCA and it had good thermal stability with $T_{5\%}$ up to 336°C. Further, OTCA@ZnO showed an excellent char-formation ability with 46.1 wt% at 700°C, more than HOTCA/ZnO with the same content of ZnO. This may be explained by the *in situ* catalytic carbonization effect of ZnO within one compound unit. Therefore, the hybridization could greatly enhance the thermo-stability and charring ability of OTCA@ZnO.

## 3.2. Flammability analysis

LOI and UL-94 results of EVA and EVA composites are presented in table 1. Obviously, EVA was extremely flammable with LOI only 17.0% and UL-94 no rating (NR). The separate introduction of APP or OTCA@ZnO showed inferior flame retardancy in EVA; whereas, the combination of APP and OTCA@ZnO enhanced the flame retardancy of EVA composites. EVA/IFR composites with APP/OTCA@ZnO in the range of 20.00/5.00 and 16.67/8.33 (wt/wt) all could reach UL-94 V-0. Meanwhile, the combustion time decreased and the melted dropping behaviour was restricted. Especially when the weight ratio of APP/OTCA@ZnO was 2 : 1, EVA/IFR-3 obtained the highest LOI (29.7%) and also passed UL-94 V-0 rating. These indicated that there was a good flame retardant synergistic effect between APP and OTCA@ZnO.

Further, the flammability analysis of EVA/APP/HOTCA/ZnO (EVA/IFR-6) and EVA/APP/HOTCA (EVA/IFR-7) with the same weight ratio of EVA/IFR-3 was also investigated in order to verify that the interaction between flame retardant and metal oxide was more efficient within one compound unit. LOI value of EVA/IFR-6 reached 27.4% and it only passed UL-94 V-2 rating, which revealed that this novel organic–metallic hybrid possessed more efficient flame retardancy than that of the physically mixing system of OTCA and nano-ZnO. The action mechanism would be further analysed subsequently. It was more obvious that EVA/IFR-7 with HOTCA as char-forming agent presented the worse flame retardant properties. Meanwhile, UL-94 V-2 rating and 27.3% LOI still could be obtained by adding only 20 wt% IFR with the ratio of APP and OTCA@ZnO maintaining at 2 : 1.

## 3.3. Fire behaviour: forced combustion

Cone calorimeter test was used to evaluate the combustion behaviour and flame characteristics of EVA composites as an effective large-scale method in this study. Several key parameters on the combustion behaviour have been collected and the corresponding data are summarized in table 3.

Figure 3 shows the heat release rate (HRR) and total heat release (THR) curves of EVA and EVA composites. Clearly, neat EVA intensely combusted after ignition, and had one sharp PHRR of 2688 kW m$^{-2}$. When adding 25 wt% APP, OTCA@ZnO, IFR-3 or IFR-6, PHRR values greatly decreased to 1358, 817, 392 and 701 kW m$^{-2}$, respectively, and the corresponding reductions were about 49.5%, 69.6%, 85.4% and 73.9% in comparison with neat EVA. Meanwhile, THR value of EVA/IFR-3 was also the lowest among the above systems and decreased to 128 MJ m$^{-2}$ from 213 MJ m$^{-2}$ for EVA with the reduction of nearly 40%.

**Table 3.** Cone data for EVA and EVA composites.

| samples | TTI (s) | $T_P$ (s) | PHRR (kW m$^{-2}$) | FPI ($10^{-2}$ s m$^2$ kW$^{-1}$) | FGI (kW m$^{-2}$ s) | THR (MJ/m$^2$) | PSPR ($10^{-2}$ m$^2$ s$^{-1}$) | TSP (m$^2$) | Peak-COP ($10^{-3}$ g s$^{-1}$) | Peak-CO$_2$P ($10^{-1}$ g s$^{-1}$) | residue (wt%) |
|---|---|---|---|---|---|---|---|---|---|---|---|
| EVA | 50 | 195 | 2688 | 1.8 | 13.8 | 213.0 | 19.8 | 14.3 | 54.9 | 15.9 | 0.5 |
| EVA/APP | 40 | 173 | 1358 | 2.9 | 7.9 | 138.6 | 14.6 | 19.9 | 14.3 | 7.2 | 16.5 |
| EVA/OTCA@ZnO | 30 | 194 | 817 | 3.6 | 4.2 | 154.8 | 10.2 | 15.8 | 7.1 | 4.7 | 4.1 |
| EVA/IFR-3 | 35 | 278 | 392 | 8.9 | 1.4 | 128.0 | 7.3 | 22.3 | 7.9 | 2.3 | 17.1 |
| EVA/IFR-6 | 30 | 225 | 701 | 4.3 | 3.1 | 135.4 | 12.5 | 24.0 | 10.4 | 3.4 | 13.3 |

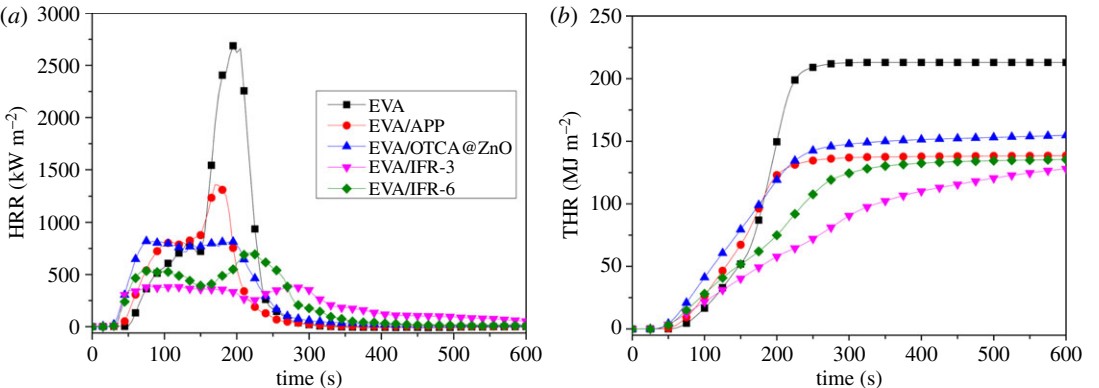

**Figure 3.** HRR (*a*) and THR (*b*) curves of pure EVA and EVA composites.

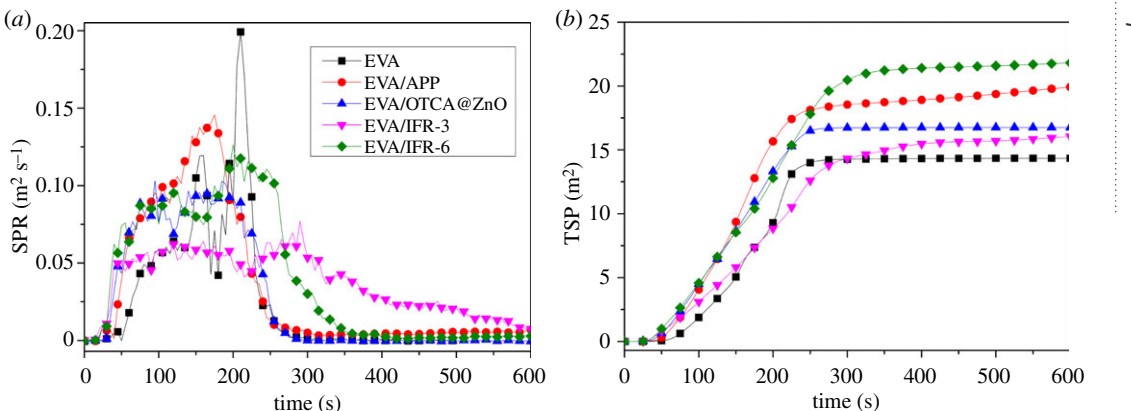

**Figure 4.** SPR (*a*) and TSP (*b*) curves of pure EVA and EVA composites.

For EVA/IFR-3 and EVA/IFR-6, both HRR curves showed two peaks, which generally corresponded to the formation and breaking processes of intumescent char layers. Yet the second peak of EVA/IFR-3 not only was much lower but also appeared obviously later than that of EVA/IFR-6. As known, ZnO could exert a flame retardant synergistic effect in IFR systems [14,15], which enhanced the strength of protective carbonaceous shield restraining the heat release during combustion, and the synergistic effect was greatly strengthened by the hybridization of ZnO for EVA/IFR-3. That is to say, the hybridization was helpful for zinc oxide to promote forming more high-quality chars by *in situ* catalytic reactions among FRs within one compound unit, and correspondingly the heat release for EVA/IFR-3 was restricted with the lowest PHRR and THR. Obviously, the physical mixture of APP/HOTCA and ZnO had lower flame retardant efficiency than the hybrid system.

From table 3, it was not difficult to find the time to ignition (TTI) with addition of APP was obviously shortened and the decrease was more obvious with the incorporation of APP/HOTCA/ZnO and APP/OTCA@ZnO. It was reported that the pyrolysis of EVA started from the deacetylation of VA monomers [27,28], and APP could accelerated the decomposition [28]. Also, ZnO regardless of the existence form could further catalyse the pyrolysis process.

The fire performance index (FPI) and the fire growth index (FGI) were investigated to more clearly understand the fire hazard of EVA composites. FPI and FGI are defined as the proportion of TTI/PHRR and the ratio of PHRR and the time to peak HRR ($T_p$), respectively. FPI relates to probability for escape in a full-scale fire situation and the larger FPI value of material means the lower fire risk [29]. For FGI, an opposite relationship appears. A smaller FGI value signifies a longer time will be taken to peak HRR, and the material possesses a lower fire danger. Seen from table 2, there were the largest FPI and lowest FGI values for EVA/IFR-3 among all the samples, and in other words, when APP/OTCA@ZnO was incorporated, the composite's burning intensity reduced and the flame growth was prohibited, undoubtedly beneficial to fire escape and fighting.

Figure 4 gives the smoke production rate (SPR) and smoke release product (TSP) curves of samples. It was found that with adding APP, OTCA@ZnO, APP/HOTCA/ZnO or APP/OTCA@ZnO into EVA, all

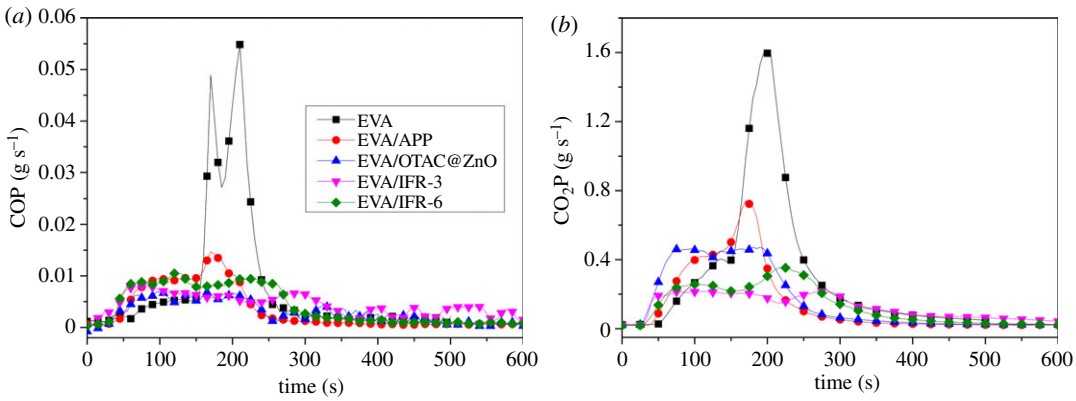

**Figure 5.** COP (*a*) and CO$_2$P (*b*) curves of pure EVA and EVA composites.

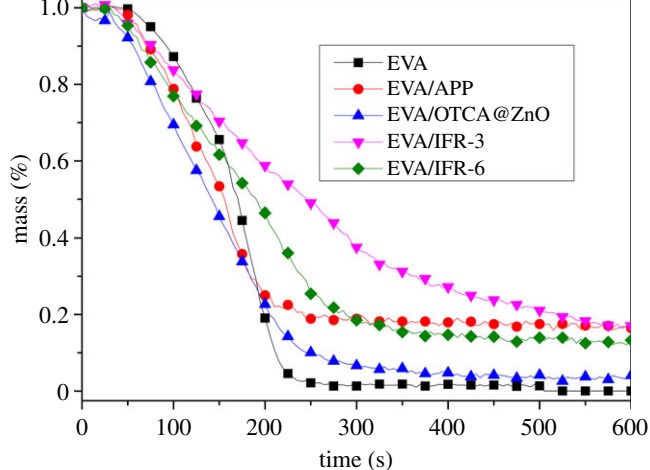

**Figure 6.** Mass loss curves of pure EVA and EVA composites.

the Peak-SPR values (PSPR) decreased compared to neat copolymer. Especially, when adding APP/ OTCA@ZnO, it reduced from 0.198 m$^2$ s$^{-1}$ for EVA to 0.073 m$^2$ s$^{-1}$, with 63.1% reduction. The results were attributed to the presence of OTCA@ZnO in IFR, which inhibited the smoke production rate. For TSP, the value of EVA/APP/OTCA@ZnO was slightly higher than neat EVA but much lower than EVA/APP, EVA/OTCA@ZnO and EVA/APP/HOTCA/ZnO, which suggested APP/OTCA@ZnO system possessed more excellent smoke suppression properties than other flame retardant systems.

The emission of toxic gas (CO) and CO$_2$ was evaluated and figure 5 shows the product curves of CO (COP) and CO$_2$ (CO$_2$P). With adding APP, OTCA@ZnO, IFR-3 or IFR-6 into EVA, all the generations of CO and CO$_2$ were restrained. Similarly, the combination of APP and OTCA@ZnO could most efficiently inhibit forming CO and CO$_2$ during combustion of composites. This also was attributed to the formation of stable char layer preventing the complete burning of polymer.

Figure 6 gives the mass loss curves, and the char residues are also listed in table 3. EVA/APP/ OTCA@ZnO lost its mass much slower than neat EVA, EVA/APP, EVA/OTCA@ZnO and EVA/APP/ HOTCA/ZnO. Meanwhile, the residue of EVA/APP/OTCA@ZnO at 600 s was 17.1 wt%, the highest value in all samples. This also was owing to the stable expansive char crust, which prohibited transmitting heat and combustible gas generated from burning as mentioned above.

Digital photographs of the residues after cone calorimeter test are displayed in figure 7. Neat EVA left nearly no residue after combustion. With the introduction of OTCA@ZnO or APP alone, sufficient amounts of residues were formed after cone test and yet the residual chars were broken up or discontinuous with no expansion, which only provided poor thermal protection. This was attributed to the result that EVA/APP composite presented inferior flame retardancy although the most residues were left in TGA tests. For EVA/IFR-3 and EVA/IFR-6, the expanded chars were formed after cone test, and yet the quantity and expansion degree of the char layer for the latter were much lower than those for the former. From figure 7, the residue of EVA/IFR-6 was thin and not strong with lots of

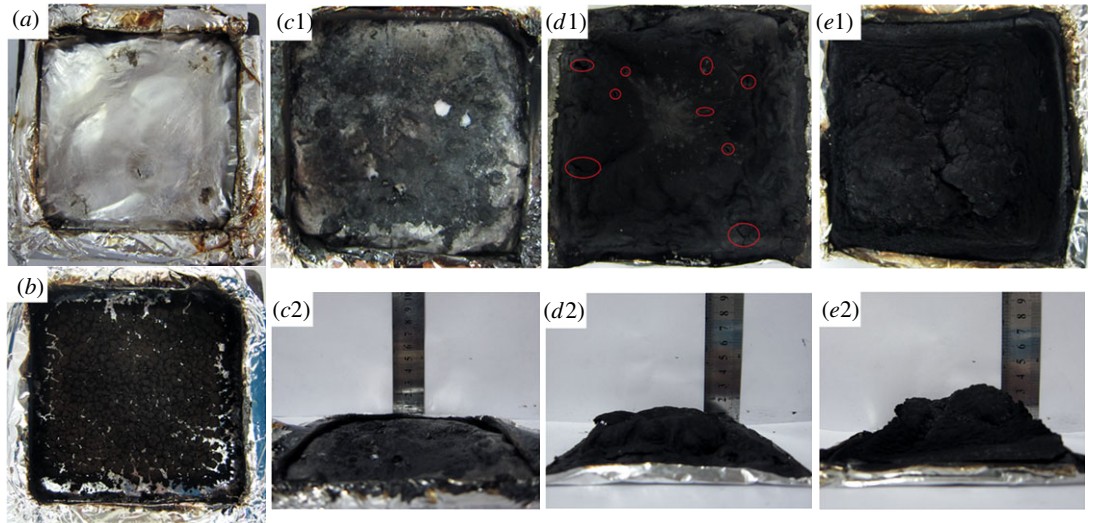

**Figure 7.** Digital photographs of residues for EVA (*a*), EVA/OTCA@ZnO (*b*), EVA/APP (*c*1,*c*2), EVA/APP/HOTCA/ZnO (*d*1,*d*2) and EVA/APP/OTCA@ZnO (*e*1,*e*2) after cone test.

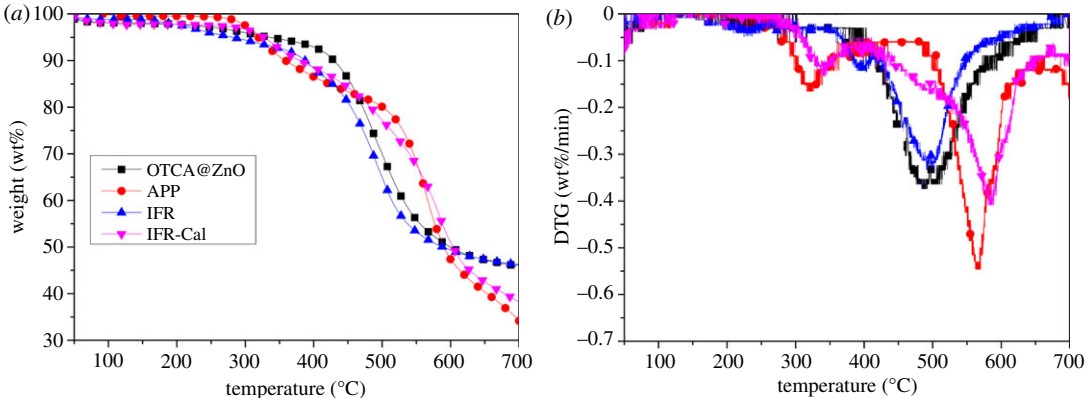

**Figure 8.** TGA (*a*) and DTG (*b*) curves of APP, IFR-3 and IFR-3 calculation under $N_2$ atmosphere.

holes and cracks on the outer surface labelled using ovals in figure 7 (*d*1), while the residue for EVA/IFR-3 was quite thick, homogeneous and strong, which proved the better carbonization effect of OTCA@ZnO than HOTCA/ZnO. The high-quality intumescent char layer could effectively prevent the transfer of heat/combustible gas, consequently improving the flame retardancy of EVA composites.

In general, all the forced combustion result analysis illuminated APP/OTCA@ZnO system could promote forming more stable intumescent char layer and obtain better flame retardant and smoke suppression properties than APP/HOTCA/ZnO system, which signified the synergistic effect of metal oxide could be more efficiently exerted within one compound unit.

## 3.4. Thermal analysis

Thermogravimetic analysis was used to rapidly evaluate the thermal degradation behaviour and charring ability of APP, OTCA@ZnO and IFR-3. TGA and DTG curves are presented in figure 8, and the data are collected in table 2. For APP, two decomposition processes appeared: the first one occurred at 319°C, generating the volatiles (mainly $NH_3$ and $H_2O$) and cross-linked polyphosphoric acids; the second one, between 480°C and 600°C, was the main decomposition process of APP [10].

When combined OTCA@ZnO with APP, it showed excellent thermal stability and char-formation ability testified by the $T_5\%$ of 310°C and the char residue of 46.3 wt% at 700°C. To further study the synergistic effect between OTCA@ZnO and APP, TGA curve of IFR-3 calculation (IFR-3-Cal) was also given in figure 8 calculated based on the following formula:

$$W_{\text{calculation}} = W_{\text{APP}} \times 67.0\% + W_{\text{OTCA@ZnO}} \times 33.5\%. \tag{3.1}$$

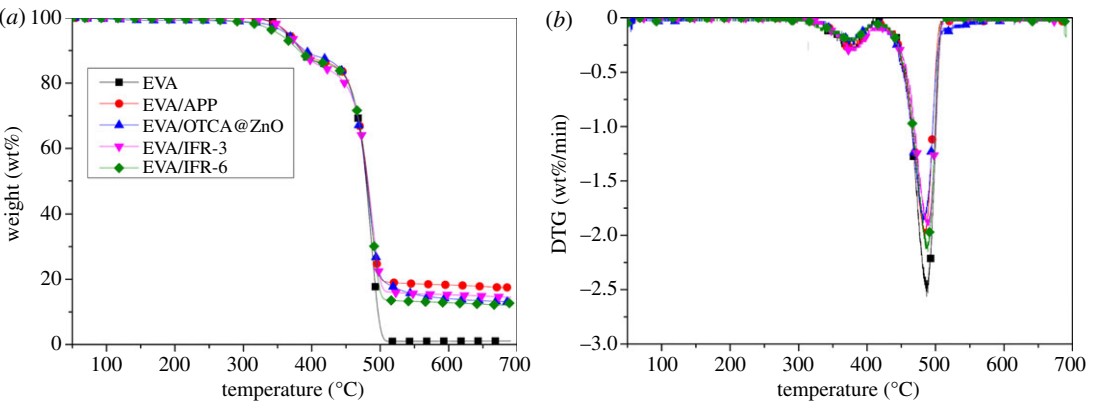

**Figure 9.** TGA (*a*) and DTG (*b*) curves of EVA and EVA composites under $N_2$ atmosphere.

**Table 4.** TGA analysis results of EVA and EVA composites.

| samples | $T_{5\%}$ (°C) | $T_{peak1}$ (°C) | $R_{peak1}$ (%/min) | $T_{peak2}$ (°C) | $R_{peak2}$ (%/min) | char residue (wt%) 500°C | 600°C | 700°C |
|---|---|---|---|---|---|---|---|---|
| EVA | 372 | 373 | 0.30 | 487 | 2.55 | 5.3 | 1.03 | 1.01 |
| EVA/APP | 364 | 377 | 0.28 | 484 | 2.01 | 20.7 | 18.3 | 17.4 |
| EVA/OTCA@ZnO | 365 | 379 | 0.25 | 483 | 1.87 | 20.8 | 13.6 | 13.1 |
| EVA/IFR-3 | 366 | 380 | 0.30 | 489 | 1.88 | 20.0 | 15.3 | 14.7 |
| EVA/IFR-6 | 354 | 375 | 0.21 | 487 | 2.11 | 16.6 | 12.8 | 12.8 |

Seen from figure 8 and table 2, the experimental $T_{5\%}$ and $T_P$ of IFR-3 both were lower than those of the theoretical values, which indicated the thermal degradation behaviours of APP/OTCA@ZnO changed as their incorporation and APP promoted the degradation of IFR. However, the charring ability of IFR-3 was greatly improved with the char residues of 46.3 wt% (700°C), while the calculation value was only 38.2 wt%. These results indicated that under higher temperatures (more than 600°C), IFR-3 expressed more excellent thermal stability and more residues were formed. It could be speculated that some chemical interactions between APP and OTCA@ZnO might take place under low temperature, generating small volatile molecules and forming cross-linking structures. Accordingly, the high-temperature degradation was restrained by the rapid formation of stable char layer promoted by the catalysis effect of ZnO. These results implied an obvious char-forming synergistic effect between OTCA@ZnO and APP, which significantly improved the charring property and thermo-stability of IFR under higher temperature.

Figure 9 shows the TGA and DTG curves of EVA and EVA composites, and the results are listed in table 4. Two decomposition processes could be seen for EVA, which were corresponding to the deacetylation step of VA monomers losing acetic acid (gas) from 320°C to 420°C and the polyene backbone decomposition between 420°C and 520°C [27–29]. The main decomposition was the second step, leaving 1.01 wt% residue at 700°C. From table 4, both $T_{5\%}$ values of EVA/IFR-3 and EVA/IFR-6 decreased compared to the initial temperature of neat EVA (372°C), especially the latter with a reduction of 18°C, which indicated that the existence form of metal oxide greatly influenced the decomposition behaviour of EVA composites. It was known that ZnO could catalyse the degradation of the matrix in IFR/polymer composites. Thus, EVA/IFR-6 with the physical blending of ZnO presented a lower initial temperature of 354°C. When zinc oxide was hybrid with organic compound, the close contact of ZnO with EVA and IFR would be restrained and so the catalytic decomposition effect of zinc oxide would not be obvious enough until the decomposition of organic components. That was to say, the hybridization could delay the catalysis decomposition effect of zinc oxide. Both EVA/IFR-3 and EVA/IFR-6 presented good char-formation abilities and still, the former generated more stable char residues under higher temperatures due to the *in situ* catalytic char-forming effect of zinc oxide in hybrid. The char residues acted as a protective shield inhibiting the outward diffusion of inflammable volatiles and heat which consequently resulted in better flame retardancy for EVA/IFR-3.

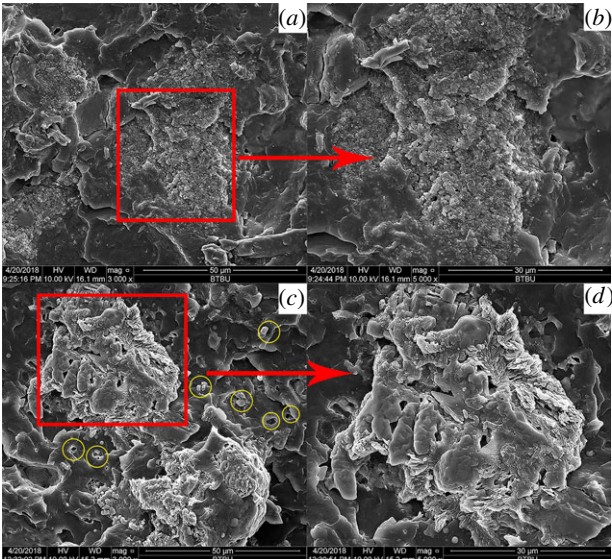

**Figure 10.** SEM images of the freeze-fractured surface EVA/IFR-3 (*a,b*) and EVA/IFR-6 (*c,d*).

## 3.5. Dispersion state and compatibility of fillers

The dispersion state and compatibility of APP/OTCA@ZnO and APP/HOTCA/ZnO (figure 10) in EVA was investigated by SEM. Comparing figure 10*a,c*, it can be seen that EVA/IFR-6 had plenty of small holes and bulges, apparently as a result of poor compatibility of pristine nano-ZnO particles with the matrix, while EVA containing OTCA@ZnO does not have this appearance. The dispersion around APP is presented in figure 10*b,d*, and it can be observed that the compatibility between the char-forming agent and APP was not bad. From figure 10*d*, it is worth noting that a large amount of nano-zinc oxide was accumulated around APP and the charring agent, and most of them existed in the form of agglomerate. Such a distribution not only deteriorated the compatibility of the fillers with the matrix, but also many nano-ZnO were not sufficiently in contact with APP and the char-forming agent, which resulted in a significant reduction of flame retardant efficiency. In contrast, the hybrid nano-zinc oxide was encapsulated by OTCA, showing a better compatibility in the matrix and APP. In this way, nano-zinc oxide together with the char-forming agent could be more uniformly dispersed in matrix, thereby improving the flame retardant efficiency. This distribution also explained why the initial decomposition temperature of EVA/IFR-3 was higher than that of EVA/IFR-6.

## 3.6. Morphology analysis of char residues

The micro-morphologies of the char layers after cone test for EVA/IFR-3 and EVA/IFR-6 composites were investigated by SEM to further analyse the action mode of *in situ* catalytic carbonization effect of OTCA@ZnO on the char-forming process of EVA composites. From figure 11*a*1,*a*2, a continuous char layer can be seen with some small holes or flaws on the surface owing to insufficient char formation during the combustion process of EVA/IFR-6. This defective char could not effectively protect the underlying material from degradation, which certainly was blamed for the weaker flame retardancy. However, for EVA/IFR-3, the char residues were obviously more compact, smooth and tight, which acted as a desired barrier from the permeation of combustion gas/oxygen and transfer of heat/mass during burning. This also was one of the primary reasons for the excellent flame retardant properties of EVA/IFR-3.

## 3.7. Structure analysis of char residues

Raman spectra are widely employed to investigate the graphitic structure and graphitization degree of char residues in the flame retardant field, and there is a certain positive correlation between the graphitization degree and the shield protection effect of char. Figure 12 presents the Raman spectra of EVA/IFR-3 and EVA/IFR-6 residues after cone test. There are two characteristic bands to be noted in the spectra: D band (approx. 1360 cm$^{-1}$) and G band (approx. 1580 cm$^{-1}$), which correspond to the

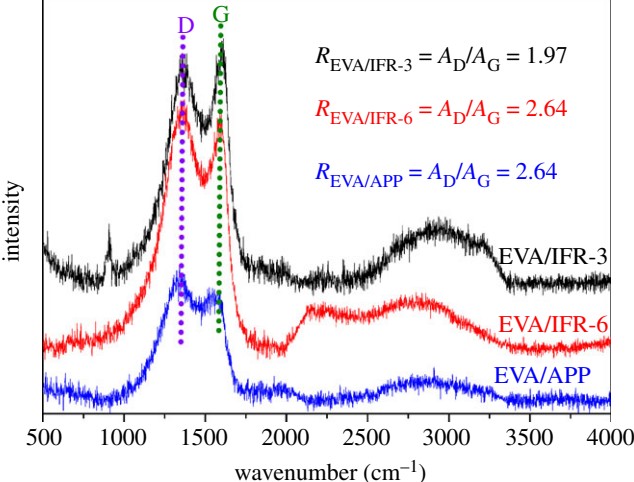

**Figure 11.** SEM images of EVA/IFR-6 (*a*1: ×500 and *a*2: ×1000) and EVA/IFR-3 (*b*1: ×500 and *b*2: ×1000) residues after cone test.

**Figure 12.** Raman spectra of EVA/IFR-3, EVA/IFR-6 and EVA/APP residues.

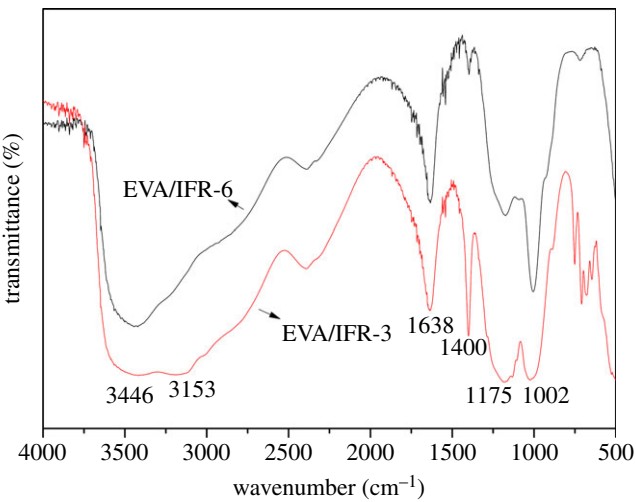

**Figure 13.** FTIR spectra of char residues for EVA/IFR-3 and EVA/IFR-6.

vibration of amorphous or unorganized and graphitic structures of carbon materials, respectively [8,30]. And, the integral peak area ratio of D band and G band, $R = A_D/A_G$, is inversely proportional to an in-plane microcrystalline size of carbon materials. The lower $R$-value indicates a higher graphitization degree of residues and the matrix will obtain a better shield protection from burning [30]. As seen from figure 12, the $R$-values of EVA/IFR-3 and EVA/IFR-6 composites were 1.97 and 2.64, which meant more graphitic char layer formed in APP/OTCA@ZnO system, and more amorphous or unorganized structures formed in APP/HOTCA/ZnO system. The high graphitic structure in EVA/IFR-3 residues more effectively prohibited the matrix from combustion, hence leading to better flame retardancy though EVA/IFR-6 also could retain enough char residues after cone test, which was in agreement with the above results. Further, EVA/APP presented a higher $R$-value (3.25), indicating that the char contained more disordered or glassy carbon structure, which explained the inferior flame retardancy of EVA/APP composite despite having the most char residues, as shown in TGA results.

## 3.8. Composition analysis of char residues

FTIR spectra of char residues after cone test for EVA/IFR-3 and EVA/IFR-6 are presented in figure 13, which could be used to investigate the chemical composition. The absorption peaks at 3450–3126 cm$^{-1}$ were ascribed to the stretching vibration of hydroxyl and amino groups, and the peak at 1638 cm$^{-1}$ belonged to C=C absorption of polyaromatic resiudues [29]. The band appearing at 1400 cm$^{-1}$ was attributed to P-N structure, and meanwhile the absorption peaks located at 1175 and 1002 cm$^{-1}$ should be the corresponding contribution of the stretching vibration of Si–O–P and C–O–P structures [31].

The above FTIR spectra revealed that there were abundant C–O–P, Si–O–P and P–N structures existing in residues, indicating that some cross-linked reactions took place between APP and OTCA@ZnO or HOTCA with zinc oxide as a kind of Lewis acid catalysing the cross-linked reactions [32], consequently phosphorus- and silicon-containing polyaromatic char residues formed during burning. Furthermore, comparing the two FTIR spectra, the characteristic peaks for EVA/IFR-3 at 1400 and 1175 cm$^{-1}$, corresponding to P–N and Si–O–P structures, showed higher intensity than EVA/IFR-6. This also explained why EVA/IFR-3 possessed higher-quality char residues as mentioned above.

## 3.9. XPS analysis of char residues

XPS spectra could be used to analyse the surface chemical structure of condensed phase [33], and the element concentrations in residues from cone test are shown in table 5. It was found that the elemental concentrations of P, O, N and Si, and the relative concentrations of P/C, O/C, N/C and Si/C for EVA/IFR-3 residue were obviously higher than those for EVA/IFR-6 residue. This suggested the hybrid compound played a key role in retaining more P, N and Si in char residues, which could effectively strengthen the char quality as discussed earlier.

**Table 5.** XPS analysis results of EVA/IFR-3 and EVA/IFR-6 residues.

| sample | C (wt%) | P (wt%) | O (wt%) | N (wt%) | Si (wt%) | Zn (wt%) | P/C (%) | O/C (%) | N/C (%) | Si/C (%) |
|---|---|---|---|---|---|---|---|---|---|---|
| EVA/IFR-3 | 23.15 | 22.65 | 48.37 | 2.16 | 1.20 | 2.47 | 97.84 | 208.94 | 9.33 | 5.18 |
| EVA/IFR-6 | 33.11 | 17.49 | 40.27 | 1.73 | 0.44 | 2.96 | 52.82 | 121.62 | 5.23 | 1.33 |

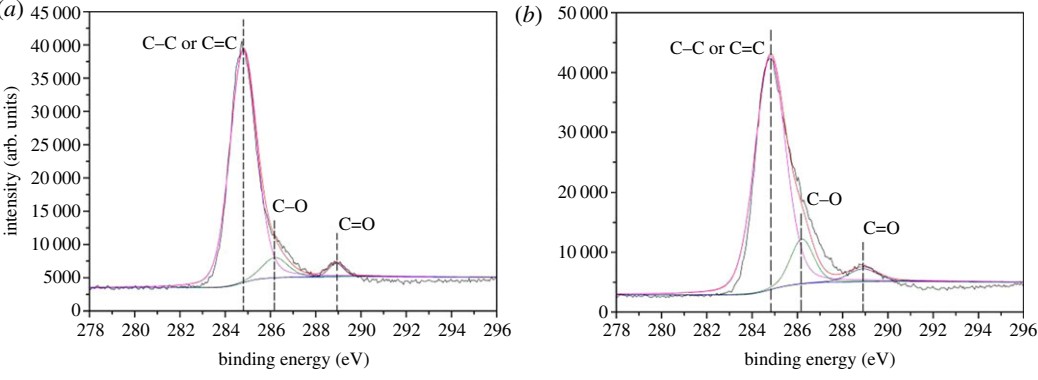

**Figure 14.** C1s spectra of char residues for EVA-6 (*a*) and EVA-3 (*b*).

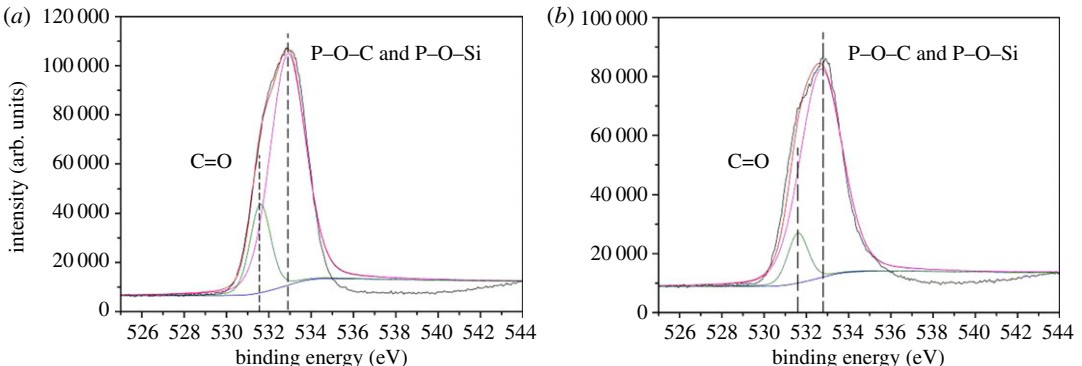

**Figure 15.** O1s spectra of char residues for EVA-6 (*a*) and EVA-3 (*b*).

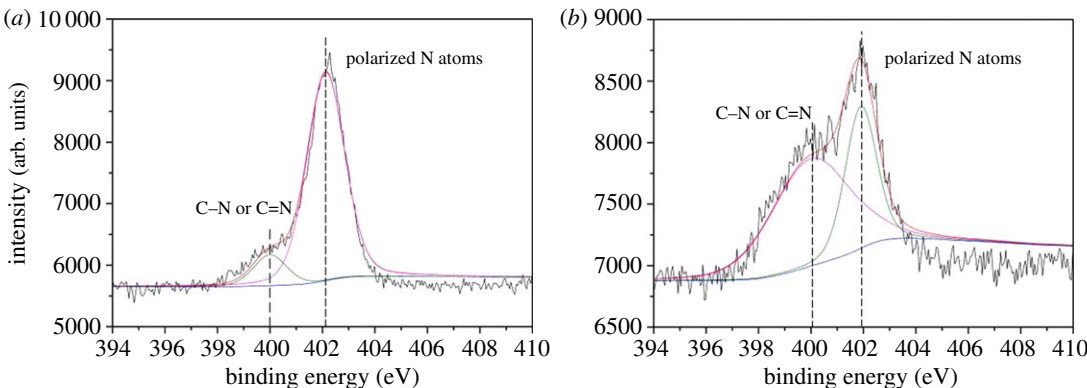

**Figure 16.** N1s spectra of char residues for EVA-6 (*a*) and EVA-3 (*b*).

Meanwhile, C1s, O1s, N1s, P2p and Si2p spectra of EVA/IFR-3 and EVA/IFR-6 residues are shown in figures 14–18 and the fitting results are listed in table 6. As seen from figure 14, the binding energy (BE) of 284.8, 286.2 and 288.9 eV for C1s peaks were designated to C–C or C=C groups in aliphatic and aromatic fragments [34], C–O in P–O–C structures and C=O groups oxidized, respectively, during combustion [8,33]. From table 6, it should be noted the percentage of C–O groups in EVA/IFR-3 residue (12.4%) was nearly two times higher than that in EVA/IFR-6 residue (7.1%), which meant the introduction of the hybrid compound could promote composites forming more cross-linking P–O–C structures. Further, for oxidized carbon atoms (C=O), the percentage in EVA/IFR-3 residue (3.0%) was lower than that in EVA/IFR-6 residue (4.4%), and the higher oxidation indicated the char barrier effect was not strong enough to restrain further oxidation of underlying matrix during combustion.

The O1s fitting curves for the two samples are shown in figure 15. The signals at around 531.6 eV showed the existence of C=O of phosphate or carbonyl groups while the BE of 533.0 eV should be

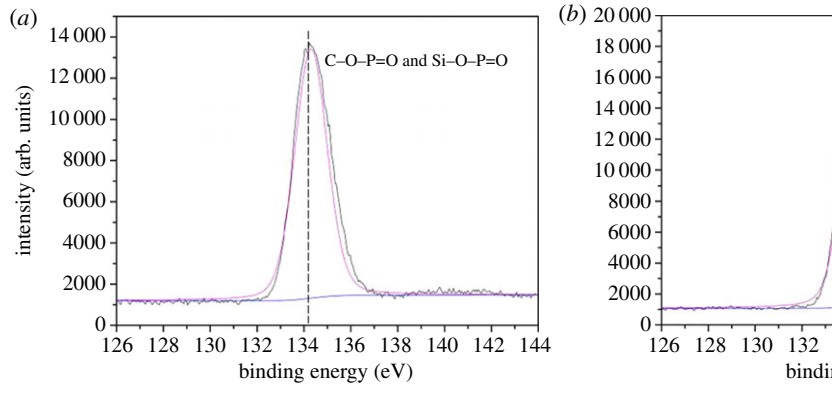

**Figure 17.** P2p spectra of char residues for EVA-6 (*a*) and EVA-3 (*b*).

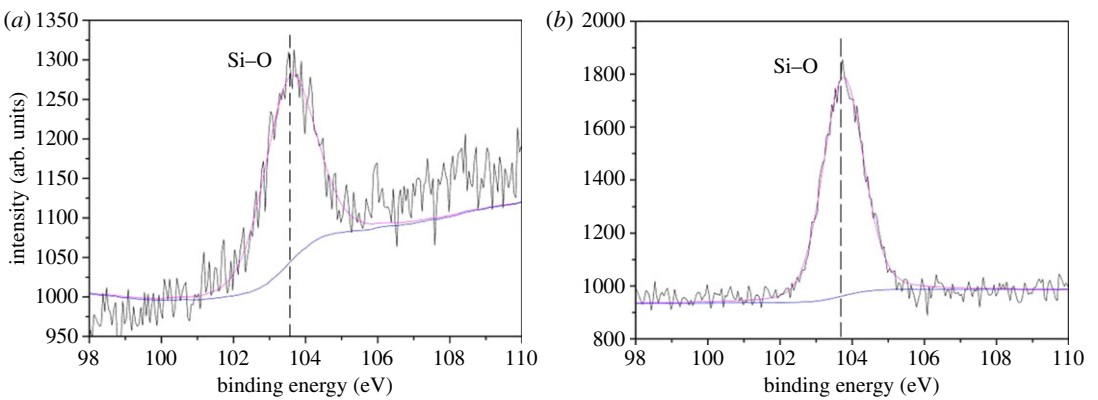

**Figure 18.** Si2p spectra of char residues for EVA-6 (*a*) and EVA-3 (*b*).

**Table 6.** Fitting results of XPS spectra for EVA/IFR-3 and EVA/IFR-6 residues.

| elements | EVA-6 | | EVA-3 | |
| --- | --- | --- | --- | --- |
| | binding energy (eV) | percentage (%) | binding energy (eV) | percentage (%) |
| C1s | 284.8 | 88.5 | 284.8 | 84.6 |
| | 286.2 | 7.1 | 286.2 | 12.4 |
| | 288.9 | 4.4 | 288.9 | 3.0 |
| O1s | 531.6 | 18.3 | 531.6 | 10.5 |
| | 532.9 | 81.7 | 532.7 | 89.5 |
| N1s | 400.0 | 11.4 | 400.0 | 34.7 |
| | 402.1 | 88.6 | 401.9 | 65.3 |
| P2p | 134.5 | 100.0 | 134.3 | 100.0 |
| Si2p | 103.7 | 100.0 | 103.6 | 100.0 |

assigned to −O− in P−O−C and P−O−Si groups [33,34]. Obviously, the higher percentage of −O− group in EVA/IFR-3 residues implied that OTCA@ZnO promoted cross-linking reactions between FRs and matrix forming more cross-linked −O− structures during combustion, which was in agreement with C1s spectra fitting results.

The N1s spectra (figure 16) at 400.0 eV corresponded to C−N or C=N in the triazine structures, and the ones at about 402.0 eV positively polarized N atoms. This suggested more N atoms existed in the form of C−N or C=N in residues of EVA/IFR-3.

As for P2p spectra, there was one peak centred at about 134.3 eV in the two residues, which corresponded to C–O–P=O and Si–O–P=O structures [34]. Yet, as reported, the P2p band in polyphosphoric acid from the decomposition of APP appeared at 132.8 eV [35]. Thus, the disappearance of the band at 132.8 eV for APP decomposition products and the shift to about 134.3 eV for EVA/IFR composites after combustion signified the interactions between APP and the charring agents, resulting in forming C–O–P=O and Si–O–P=O structures. It may be deduced that the interaction in EVA/IFR-3 was more efficient than in EVA/IFR-6 because of the much higher content of P in residues (table 5) which came from the polyphosphoric acid evolved into residues in the char-formation process. As seen from the Si2p spectra of chars (figure 18), the binding energy at around 103.6 eV was assigned to Si–O bonds in Si–O–P or Si–O–Si. And the existence of Si improved the residue strength and thus helped the formation an excellent protective barrier.

Therefore, the above results suggested that the more efficient interactions in APP/OTCA@ZnO system promoted the formation of richer P–O–C, P–N and P–O–Si cross-linked structures in char-forming process, which facilitated the improvement of the strength and barrier properties of residues.

# 4. Conclusion

A novel hybrid OTCA@ZnO was prepared and the hybridization obviously improved the charring ability and dispersity of OTCA@ZnO. The combination of APP and OTCA@ZnO showed higher efficiency in enhancing flame retardancy and smoke suppression functions than APP/HOTCA/ZnO system in EVA. When 25 wt% APP/OTCA@ZnO with the optimal ratio 2/1 was introduced, the composite (EVA/IFR-3) obtained the highest LOI value of 29.7% and UL-94 V-0 rating. APP/OTCA@ZnO influenced the combustion behaviour of EVA composites, resulting in lower HRR, THR, FGI, SPR, TSR, COP and $CO_2P$ values than APP/HOTCA/ZnO system. This could be explained by the more efficient interactions between IFR and zinc oxide within one compound unit. Thermal analysis results presented an obvious synergistic effect in APP/OTCA@ZnO system, strengthening efficiently the charring capacity of the hybrid system. The char residue reached 46.3 wt% at 700°C, while 38.2 wt% residues were obtained by calculation. Further, the residue investigation indicated APP/OTCA@ZnO system could form more compact and tight graphitic char layer containing richer C–O–P, P–N and Si–O–P cross-linked structures due to the *in situ* catalytic carbonization effect of hybrid. Just the high-quality char layer brought more efficient suppression of combustion and smoke release for EVA composites. The organic–metallic hybrid may be a promising route for flame retardant polymer materials.

Data accessibility. Data supporting this article are provided as the electronic supplementary material.

Authors' contributions. B.X. designed the study, ran the data analysis and wrote the manuscript; W.M. and L.S. carried out the whole laboratory work; L.Q. advised on the design of the study and provided technology assistance; Y.Q. participated in the data analysis. All authors gave final approval for the publication.

Competing interests. We declare we have no competing interests.

Funding. This work has been supported by the Beijing Municipal Natural Science Foundation (grant no. 2192014).

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
