## [Reviewer comments · Royal Society Open Science]

Review History

RSOS-181413.R0 (Original submission)

Review form: Reviewer 1

Is the manuscript scientifically sound in its present form?

No

Are the interpretations and conclusions justified by the results?

Yes

Is the language acceptable?

No

Is it clear how to access all supporting data?

Not Applicable

Do you have any ethical concerns with this paper?

No

Have you any concerns about statistical analyses in this paper?

Yes

Recommendation?

Major revision is needed (please make suggestions in comments)

Comments to the Author(s)

This study reported the use of organic-metallic hybrid charring material as a flame retardant. Although the authors have done a lot of characterization work, the current version would need some revisions before it can be considered for publication. Some comments as below:

-Abbreviations should not be used in the title.

-The authors claimed it is a metallic hybrid, but what they used is ZnO which is an oxide. This is very misleading.

-Many abbreviations such as SCTCA are not specified.

-The TEM images of Fig. 1 do not look good. The ZnO nanoparticles look more like aggregates. The SCTCA@ZnO looks entirely different than ZnO with different electron densities. However, SCTCA modification shouldn't change the TEM contrast of ZnO.

-The TGA curves in Fig. 8 do not really have any significant differences between samples. The minor difference could be just due to experimental variations.

-A lot of the measurements in the study are based on one-time measurement which is a potential issue.

-Each fitted peak in XPS spectra should be labeled.

-There are a lot of grammar mistakes and typos, even in the title of the manuscript. This would really need a lot of work on it.

-Besides the smoke production and release. Some other performance characterization as a flame retardant should be added, especially given that the mass loss seems very marginal without statistical analysis.

Review form: Reviewer 2**Is the manuscript scientifically sound in its present form?**

No

Are the interpretations and conclusions justified by the results?

Yes

Is the language acceptable?

Yes

Is it clear how to access all supporting data?

Yes

Do you have any ethical concerns with this paper?

No

Have you any concerns about statistical analyses in this paper?

I do not feel qualified to assess the statistics

Recommendation?

Major revision is needed (please make suggestions in comments)

Comments to the Author(s)

1. In abstract, page 2 line 40-42. The authors mentioned the comparison of the EVA/APP/SCTCA@ZnO and EVA/APP/SCTCA/ZnO systems, but in the subsequent text, they used hydrolyzed product HSCTCA/ZnO. Why did the authors use the hydrolyzed product HSCTCA/ZnO as a control rather than the compound of SCTCA with ZnO?
2. In introduction, extensive literature regarding current progress on the flame retardants EVA composites should be included. In addition, the research progress about ZnO as flame retardant of polymer should be reviewed.
3. Clarify the novelty of this study, compared with other similar research.
4. Clarify SCTCA.
5. How to determine the feeding ratio for the preparation of SCTCA?
6. Describe the reaction mechanism for the preparation of SCTCA with a schematic.
7. 2.2, what was used to adjust pH value?
8. P4, "Scheme 1 presented the routes of synthesis for SCTCA and SCTCA@ZnO", but the route for SCTCA synthesis was missed.
9. The full names for all abbreviations should be given when they appear for the first time, such as TGA, LRS and PHRR.
10. P7, what are HSCTCFA, HSCTFA/ZnO and SCTCFA-ZnO? They are not consistent with the expressions in the text and table! Please use the same expressions in full text, figures and tables!
11. Page 3 line 38-42 "Firstly, 20 mL DMAc solution containing 32 g γ -aminopropyltriethoxy silane was added into a three-necked flask equipped with a mechanical stirrer. Then, 27 g CYC and 14.6 g TEA in 20 mL DMAc was slowly added dropwise with vigorously stirring in an external ice bath". Was it ice bath or ice-water bath? What was the temperature? In addition, the reaction temperature was gone up to 50 °C after three hours. If ice bath was used, how to increase the temperature? The description is too confusing.
12. Page 3 line 51-52 "After dried to a constant weight under vacuum, the intermediate SCTCA (light yellow powder) was obtained". What was the drying temperature?
13. Page 4 Line 3-4, what is the purpose for activation of nano ZnO? Is there any reference?
14. Page 4 line 10, "The solid was washed three times with hot water", why was hot water used?
15. The authors should characterize the dispersion of the fillers in the composites by SEM or TEM.

Decision letter (RSOS-181413.R0)

11-Dec-2018

Dear Dr Xu:

Title: Enhancement of an organic-metallic hybrid charring agent on flame retardancy of EVA
 Manuscript ID: RSOS-181413

The editor assigned to your manuscript has now received comments from reviewers. We would like you to revise your paper in accordance with the referee and Subject Editor suggestions which can be found below (not including confidential reports to the Editor). Please note this decision does not guarantee eventual acceptance.

Please submit your revised paper before 03-Jan-2019. Please note that the revision deadline will expire at 00.00am on this date. If we do not hear from you within this time then it will be assumed that the paper has been withdrawn. In exceptional circumstances, extensions may be possible if agreed with the Editorial Office in advance. We do not allow multiple rounds of revision so we urge you to make every effort to fully address all of the comments at this stage. If deemed necessary by the Editors, your manuscript will be sent back to one or more of the original reviewers for assessment. If the original reviewers are not available we may invite new reviewers.

Please also include the following statements alongside the other end statements. As we cannot publish your manuscript without these end statements included, if you feel that a given heading is not relevant to your paper, please nevertheless include the heading and explicitly state that it is not relevant to your work.

- Ethics statement

Please clarify whether you received ethical approval from a local ethics committee to carry out your study. If so please include details of this, including the name of the committee that gave consent in a Research Ethics section after your main text. Please also clarify whether you received informed consent for the participants to participate in the study and state this in your Research Ethics section.

OR

Please clarify whether you obtained the necessary licences and approvals from your institutional animal ethics committee before conducting your research. Please provide details of these licences and approvals in an Animal Ethics section after your main text.

OR

Please clarify whether you obtained the appropriate permissions and licences to conduct the fieldwork detailed in your study. Please provide details of these in your methods section.

- Acknowledgements

Once again, thank you for submitting your manuscript to Royal Society Open Science and I look

forward to receiving your revision. If you have any questions at all, please do not hesitate to get in touch.

On behalf of the Subject Editor Professor Anthony Stace and the Associate Editor Professor Eva Hevia.

RSC Associate Editor:
Comments to the Author:
(There are no comments.)

RSC Subject Editor:
Comments to the Author:
(There are no comments.)

Reviewers' Comments to Author:
Reviewer: 1

Comments to the Author(s)

This study reported the use of organic-metallic hybrid charring material as a flame retardant. Although the authors have done a lot of characterization work, the current version would need some revisions before it can be considered for publication. Some comments as below:

-Abbreviations should not be used in the title.

-The authors claimed it is a metallic hybrid, but what they used is ZnO which is an oxide. This is very misleading.

-Many abbreviations such as SCTCA are not specified.

-The TEM images of Fig. 1 do not look good. The ZnO nanoparticles look more like aggregates. The SCTCA@ZnO looks entirely different than ZnO with different electron densities. However, SCTCA modification shouldn't change the TEM contrast of ZnO.

-The TGA curves in Fig. 8 do not really have any significant differences between samples. The minor difference could be just due to experimental variations.

-A lot of the measurements in the study are based on one-time measurement which is a potential issue.

-Each fitted peak in XPS spectra should be labeled.

-There are a lot of grammar mistakes and typos, even in the title of the manuscript. This would really need a lot of work on it.

-Besides the smoke production and release. Some other performance characterization as a flame retardant should be added, especially given that the mass loss seems very marginal without statistical analysis.

Reviewer: 2

Comments to the Author(s)

1. In abstract, page 2 line 40-42. The authors mentioned the comparison of the EVA/APP/SCTCA@ZnO and EVA/APP/SCTCA/ZnO systems, but in the subsequent text, they used hydrolyzed product HSCTCA/ZnO. Why did the authors use the hydrolyzed product HSCTCA/ZnO as a control rather than the compound of SCTCA with ZnO?
2. In introduction, extensive literature regarding current progress on the flame retardants EVA composites should be included. In addition, the research progress about ZnO as flame retardant of polymer should be reviewed.
3. Clarify the novelty of this study, compared with other similar research.
4. Clarify SCTCA.
5. How to determine the feeding ratio for the preparation of SCTCA?
6. Describe the reaction mechanism for the preparation of SCTCA with a schematic.
7. 2.2, what was used to adjust pH value?
8. P4, "Scheme 1 presented the routes of synthesis for SCTCA and SCTCA@ZnO", but the route for SCTCA synthesis was missed.
9. The full names for all abbreviations should be given when they appear for the first time, such as TGA, LRS and PHRR.
10. P7, what are HSCTCFA, HSCTFA/ZnO and SCTCFA-ZnO? They are not consistent with the expressions in the text and table! Please use the same expressions in full text, figures and tables!
11. Page 3 line 38-42 "Firstly, 20 mL DMAc solution containing 32 g γ -aminopropyltriethoxy silane was added into a three-necked flask equipped with a mechanical stirrer. Then, 27 g CYC and 14.6 g TEA in 20 mL DMAc was slowly added dropwise with vigorously stirring in an external ice bath". Was it ice bath or ice-water bath? What was the temperature? In addition, the reaction temperature was gone up to 50 °C after three hours. If ice bath was used, how to increase the temperature? The description is too confusing.
12. Page 3 line 51-52 "After dried to a constant weight under vacuum, the intermediate SCTCA (light yellow powder) was obtained". What was the drying temperature?
13. Page 4 Line 3-4, what is the purpose for activation of nano ZnO? Is there any reference?
14. Page 4 line 10, "The solid was washed three times with hot water", why was hot water used?
15. The authors should characterize the dispersion of the fillers in the composites by SEM or TEM.

Author's Response to Decision Letter for (RSOS-181413.R0)

See Appendix A.

RSOS-181413.R1 (Revision)

Review form: Reviewer 1

Is the manuscript scientifically sound in its present form?

Yes

Are the interpretations and conclusions justified by the results?

Yes

Is the language acceptable?

Yes

Is it clear how to access all supporting data?

Yes

Do you have any ethical concerns with this paper?

No

Have you any concerns about statistical analyses in this paper?

No

Recommendation?

Accept as is

Comments to the Author(s)

The authors have responded to the previous concerns and the reviewer does not have additional comments except that the authors used the phrase "organic-metallic hybrid". Is this on purpose to be distinct from metal organic framework (MOF)? The term as used by the authors is not frequently used.

Decision letter (RSOS-181413.R1)

08-Feb-2019

Dear Dr Xu:

Title: Enhancement of an organic-metallic hybrid charring agent on flame retardacy of ethylene-vinyl acetate copolymer

Manuscript ID: RSOS-181413.R1

It is a pleasure to accept your manuscript in its current form for publication in Royal Society Open Science. The chemistry content of Royal Society Open Science is published in collaboration with the Royal Society of Chemistry.

On behalf of the Subject Editor Professor Anthony Stace and the Associate Editor Professor Eva Hevia.

RSC Associate Editor:
Comments to the Author:
(There are no comments.)

RSC Subject Editor:
Comments to the Author:
(There are no comments.)

Reviewer(s)' Comments to Author:
Reviewer: 1

Comments to the Author(s)

The authors have responded to the previous concerns and the reviewer does not have additional comments except that the authors used the phrase "organic-metallic hybrid". Is this on purpose to be distinct from metal organic framework (MOF)? The term as used by the authors is not frequently used.

Appendix A

Dear Editors,

Thank you for your letter and the reviewers' comments concerning our manuscript entitled "Enhancement of an organic-metallic hybrid charring agent on flame retardancy of EVA". Those comments are all valuable and very helpful for revising and improving our paper, as well as the important guiding significance to our researches. We have uploaded a revised manuscript plus responses to the reviewers' comments that we feel addresses adequately all the issues proposed. Words in **green** are the changes for the comments of **Referees** in the modified manuscript. The corrections and responses to the reviewer's comments are as following:

Responses to the reviewers' comments:

Reviewer: 1

Comment 1: Abbreviations should not be used in the title.

Response: Thanks for the reviewer's kind advice. According to the reviewer's suggestion, the title has been changed to "Enhancement of an organic-metallic hybrid charring agent on flame retardancy of ethylene-vinyl acetate copolymer" in the modified manuscript.

Comment 2: The authors claimed it is a metallic hybrid, but what they used is ZnO which is an oxide. This is very misleading.

Response: Thank you very much for your suggestions. Based on the definition, hybrid materials are composites consisting of two constituents at the nanometer or molecular level. Commonly one of these compounds is inorganic (including inorganic oxide, metal oxide, etc) and the other one organic in nature and they are connected by the covalent bonds. The synthetic idea of hybrid polymer materials is that if there are groups on the polymer chain that can participate in the hydrolysis and condensation process, through the hydrolysis and condensation of these functional groups with inorganic precursors, the organic polymer and inorganic hybrid polymer materials can be formed by covalent bonding. Thus, they differ from traditional composites where the constituents are at the macroscopic (micrometer to millimeter) level. Mixing at the microscopic scale leads to a more homogeneous material that either shows characteristics in the two original phases or even new properties. Specific to this work, nano-ZnO as a nanometer scale inorganic metal oxide was modified by the triazine oligomer containing functional groups of silicon ethoxy

and the two could be connected by the covalent bonds generated through the hydrolysis and condensation of these functional groups with hydroxyl on nano-ZnO, forming a nanometer scale composite. Based on this, the composite in this work was regarded as metallic hybrid material.

At the same time, we have also consulted some previous works and found some metallic hybrids with metal oxides as raw materials. The references are given as examples:

1. Shankar S, Oun A A, Rhim JW. 2018 Preparation of antimicrobial hybrid nano-materials using regenerated cellulose and metallic nanoparticles. *Int J Biol Macromol.* **107**, 17-27. (doi:10.1016/j.ijbiomac.2017.08.129)

2. Chae S, Ji BG, Kim A, Choi E, Kwon SH, Zheng LS, Kang K, Nam M, Paik T, Kim KS, Pyo SG. 2017 Improving Photocatalytic Antibacterial Activity from the Facile Fabrication ZnO Nano Particles with Metallic Hybrid-MWCNT. *J. Nanosci. Nanotechnol.* **17**, 3496–3499. (doi:10.1166/jnn.2017.14089)

3. Gautier J, Allard-Vannier E, Hervé-Aubert K, Soucé M, Chourpa I. 2013 Design strategies of hybrid metallic nanoparticles for

theragnostic applications. *Nanotechnology*. **24**, 432002.

(doi:10.1088/0957-4484/24/43/432002)

Comment 3: Many abbreviations such as SCTCA are not specified.

Response: Thanks for the reviewer's suggestion. According to your suggestion, we have identified the abbreviations of the flame retardants again:

OTCA: organic triazine charring agent.

HOTCA: hydrolyzed OTCA.

OTCA@ZnO: organic triazine charring agent hybrid with zinc oxide.

HOTCA/ZnO: physical blending of HOTCA and nano-ZnO.

Comment 4: The TEM images of Fig. 1 do not look good. The ZnO nanoparticles look more like aggregates. The SCTCA@ZnO looks entirely different than ZnO with different electron densities. However, SCTCA modification shouldn't change the TEM contrast of ZnO.

Response: Thanks for your comment. As seen from Fig. 1(a), nano-ZnO possessed shapes like long rods and strongly aggregated

into clusters of several micrometers in size due to the high surface energy and lots of hydroxyl groups on the surface. The average particle size of 80 nm × 15 nm of nano-ZnO and yet the aggregation was measured by a 500 nm scale. In fact, the aggregation for nanometer metal oxide particles commonly appears due to the high surface energy and they will agglomerate into clusters much larger than 500 nm rather than being dispersed at a particle size of nanometer each, which will strongly affect the dispersion and distribution of particles in the matrix. Thus the organic surface modification of nanometer particles may be an optimized way to obstruct the aggregation of nanometer particles. Just for this reason, this paper hybridizes nano-zinc oxide with organic charring agent in order to improve the dispersion of nano-zinc oxide, which is the primary goal of this work.

In fact, the goals of TEM tests were to identify the influence of hybridization on the dispersion of nano-ZnO, though the contrast difference of the test results may be because the operators are different twice, responded by the testing organization for TEM images in Figure 1. From Figure 1(a, b), the contrast of particles' morphology and size

before and after hybridization of nano-ZnO was enough obvious. OTCA@ZnO showed a distribution like “seeds” in “melons”, and we selected a seed to enlarge, as shown in Figure 1(c). It was not difficult to see from the enlarged Figure 1(c) that although the nano-zinc oxide still agglomerated, its particle size had been reduced from much more than 500 nm to about 100 nm, indicating that the dispersion of nano-zinc oxide after hybridization was improved in organic char-forming agent. This may be explained by that the hybridization consumed the surface hydroxyl groups through the condensation reactions and lowered the surface energy of nano-ZnO. The organic modifier layer covering on surface of nano-ZnO generated the large steric hindrance, obstructing the aggregation of particles. However, unlike the seeds in melons, the inorganic and organic interfaces are not so clear, because the hydroxyl groups on the surface of the nano-zinc oxide and the hydroxyl groups on the OTCA undergo condensation reaction, which improves the interfacial compatibility of the two, and at the same time, most of particles of OTCA@ZnO showed darker center and lighter edger, which was typical of hybridization structure of inorganic nanometer particles.

Comment 5: The TGA curves in Fig. 8 do not really have any significant differences between samples. The minor difference could be just due to experimental variations.

Response: Thanks. Your observation is very careful and indeed, some of the curves seemed little significant difference, such as the curves between OTCA@ZnO and IFR, APP and IFR-Cal. In fact, each test has been carried out more than once in the experimental process to eliminate potential errors. For this TGA section, our primary goal was to demonstrate synergy effect between OTCA@ZnO and APP, which could be seen from the two curves of IFR and IFR-Cal. The experimental $T_{5\%}$ and T_p of IFR-3 both were lower than those of the theoretical values, which indicated the thermal degradation behaviors of APP/OTCA@ZnO changed as their incorporation and APP promoted the degradation of IFR. Further, IFR-3 with more residues had better charring ability than the calculation value especially in high temperature.

First of all, the two curves of OTCA@ZnO and IFR seemed close to each other, but the initial decomposition temperatures were significantly different. This indicated that the synergy of OTCA@ZnO

and APP made IFR decompose earlier. In fact, the contrast of the two curves of OTCA@ZnO and IFR was of no significance, and it was also normal for the char residues of both to be similar, even if the char residues of OTCA@ZnO was higher or lower than that of IFR, because the amount of char was not the exclusive crucial factor on the properties of flame retardants.

Secondly, for the two curves of APP and IFR-Cal, it could be clearly seen that the calculated IFR-Cal was closer to the thermal weight loss process of APP. As you said, the two curves are almost the same and it was really easy to be regarded as certain experimental errors. In fact, this was because APP accounted for a large part (2/3) of IFR, so that the calculated theoretical curve was very close to APP curve. On the contrary, it was the fact that the two almost identical curves proved the addition of OTCA@ZnO had greatly demonstrated the synergy effect between OTCA@ZnO and APP, rather than simply mixing.

Comment 6: A lot of the measurements in the study are based on one-time measurement which is a potential issue.

Response: Thank you very much for your suggestions. For the

experiments in this work, more than once tests have been carried out for each formula in parallel. The principle insisted on was that each formula was tested twice, and if the two results were the same or there were only very small differences for each samples, it was considered reasonable and adopted. On the contrary, if the results of the two repeated tests were significantly different, the third test would be executed until repeatable results were obtained. At the same time, the tests were in full compliance with international standards. For example, a calorimeter was used to test two samples per formula. In addition, every test was done according to its standards. Limited oxygen index (LOI) values were measured according to American Society for Testing and Materials (ASTM) D2863-17. UL-94 vertical burning rating tests were performed following the procedures in ASTM D3801-10. The forced combustion behavior was measured according to ISO 5660-1. In a word, the purpose of these was to reduce the error caused by the accidental experiment, ensuring the objectivity and accuracy of the experimental results.

Comment 7: Each fitted peak in XPS spectra should be labeled.

Response: Thanks for the reviewer's suggestion. According to

your suggestion, each fitted peak in XPS spectra was labeled as follows.

Figure 14. C1s spectra of char residues for EVA-6 (a) and EVA-3 (b).

Figure 15. O1s spectra of char residues for EVA-6 (a) and EVA-3 (b).

Figure 16. N1s spectra of char residues for EVA-6 (a) and EVA-3 (b).

Figure 17. P2p spectra of char residues for EVA-6 (a) and EVA-3 (b).

Figure 18. Si2p spectra of char residues for EVA-6 (a) and EVA-3 (b).

Comment 8: There are a lot of grammar mistakes and typos, even in the title of the manuscript. This would really need a lot of work on it.

Response: Thanks for the referee's kind advice. We have invited a native English speaker to read this manuscript and corrected the grammar mistakes, which were marked in green in the revised manuscript.

Comment 9: Besides the smoke production and release. Some other performance characterization as a flame retardant should be added, especially given that the mass loss seems very marginal without statistical analysis.

Response: Thanks for the referee's kind advice. In this work, the flame retardancy and thermal behaviour of intumescent flame retardant EVA composites combining OTCA@ZnO and ammonium polyphosphate (APP) were investigated using limited oxygen index (LOI), UL-94 vertical burning, cone calorimetry and TGA. The structure and morphology of chars were investigated by scanning electron microscopy (SEM), FTIR, laser Raman spectroscopy analysis (LRS) and X-ray photoelectron spectroscopy (XPS). Results revealed that the flame retardancy and smoke suppression properties of EVA were significantly improved by introducing APP/OTCA@ZnO. TGA results indicated that APP/OTCA@ZnO presented an excellent synergistic effect and promoted the char formation of EVA composites. Residue analysis results showed more char with high quality connected by richer P-O-C, P-N and P-O-Si structures was formed in APP/OTCA@ZnO system than APP/HOTCA/ZnO system, which

consequently suppressed more efficiently the combustion and smoke production due to the in situ catalytic carbonization effect of hybrid. For the flame retardant, the organic triazine charring agent hybrid with zinc oxide (OTCA@ZnO) was well characterized through fourier transform infrared spectrometry (FTIR), solid state nuclear magnetic resonance (SSNMR), transmission electron microscopy (TEM) and thermogravimetric analysis (TGA). Results revealed that OTCA@ZnO exhibited excellent thermal stability and dispersity after hybridization. For an IFR system, commonly these tests could well characterize the flame retardant properties and the properties of the flame retardants. Lots of previous works also referred these characterizations and maybe the performance characterizations in previous references were less than those in this work. If you insist on still lacking performance characterizations, please kindly tell us the specific methods, which would be carried out in the future.

Thank you very much for your comments and suggestions again.

Reviewer: 2

Based on the other reviewer' suggestions, the abbreviations appearing below: SCTCA, HSCTCA, SCTCA@ZnO and

SCTCA/ZnO have been replaced by the modified ones: OTCA, HOTCA, OTCA@ZnO and HOTCA/ZnO, respectively.

Comment 1: In abstract, page 2 line 40-42. The authors mentioned the comparison of the EVA/APP/SCTCA@ZnO and EVA/APP/SCTCA/ZnO systems, but in the subsequent text, they used hydrolyzed product HSCTCA/ZnO. Why did the authors use the hydrolyzed product HSCTCA/ZnO as a control rather than the compound of SCTCA with ZnO?

Response: Thank you very much for your question. In fact, this issue has been considered in this experiment, and the reasons for using the hydrolyzed product HOTCA/ZnO (HSCTCA/ZnO) as a control rather than OTCA/ZnO (SCTCA/ZnO) were as follows:

From the synthesis process of OTCA@ZnO seen in Scheme 2, the ethoxy groups of γ -aminopropyltriethoxy silane became firstly hydroxyl groups after hydrolysis. Next, a part of the hydroxyl groups of the OTCA after hydrolysis condensed with the hydroxyl groups on the surface of nano-ZnO, and the other part of hydroxyl groups self-condensed. Further, the hydrolysis product of OTCA named as HOTCA was prepared following the same procedure except for

without nano-ZnO, and HOTCA physically blended with nano-ZnO would be used as a comparative object of the hybrid OTCA@ZnO. In order to get closer to the real situation of the compound, the hydrolyzed product HOTCA, physically blending with nano-ZnO (HOTCA/ZnO), was used as a control rather than the mixture of OTCA with nano-ZnO, because the network structure after self-condensation of hydroxyl groups was more similar to that of OTCA@ZnO.

In addition, the flame retardant properties of EVA/APP/OTCA/ZnO composite with the same addition amount and ratio of EVA/APP/HOTCA/ZnO were investigated in the experiment process, and unexpectedly, the properties were less than those of EVA/APP/HOTCA/ZnO composite containing the hydrolyzed product HOTCA, not to mention EVA/APP/OTCA@ZnO. Meantime, the thermal properties of OTCA were also measured, and yet it was found both the thermal stability and char forming ability were inferior to the hydrolyzed product HOTCA.

Thus, considering the above issues, the hydrolyzed product HOTCA blending with nano-ZnO (HOTCA/ZnO) rather than

OTCA/ZnO was used as a comparative object of the hybrid OTCA@ZnO.

Comment 2: In introduction, extensive literature regarding current progress on the flame retardants EVA composites should be included. In addition, the research progress about ZnO as flame retardant of polymer should be reviewed.

Response: Thanks for the referee's kind advice. We have added the current progress on the flame retardants EVA composites and the research progress about ZnO as flame retardant of polymer into introduction, which were labeled in green, and the according references were added in the revised manuscript.

Comment 3: Clarify the novelty of this study, compared with other similar research.

Response: At present, lots of flame retardant synergist, such as ZnO, have been widely studied to enhance the efficiency of IFR systems on polymers. It was reported that a small amount of metal oxides (e.g. ZnO and La_2O_3) could significantly improve the thermal stability and flame retardancy of IFR composites. However, the

existing works showed metal oxides were usually blended physically with IFR in polymers and thus it was difficult for the mixtures to fully show their synergism due to the very low loading amounts and serious reunion of oxides. The final properties of polymeric composites strongly depend on particles' dispersion and distribution in the matrix. Moreover, the weak compatibility of metal oxides in composites also was a severe disadvantage, which deteriorated the physical and mechanical properties. Although the synergistic effect of the metal oxide and the conventional flame retardant improved the flame retardancy of materials, the polarity of the inorganic metal compound was largely different from that of the polymer matrix and easily agglomerated, if it was physically added. So, the organic surface modification of metal oxides, especially employing those modifiers with fire resistance, may be a prospective method, which would not only settle the above matters, but also bring more efficient synergistic effect of metal oxides within one compound unit with flame retardants.

Silicon-containing compounds are types of environment-friendly and highly efficient flame retardants which could form a protective carbonaceous layer with excellent barrier effect at high temperature. It

was reported that silicone could comparatively lower the heat release rate and release product of toxic gas (CO), and the synergistic effect was especially obvious between silicone and IFR with the protective char layer containing Si-O and Si-C bonds. Further, silicones could improve the interfacial compatibility of fillers and matrix by chemical interactions, which regulated the structure and property of composites.

In this work, based on the flame retardant effectiveness of triazine compounds and silicon, and the synergistic effect of ZnO, a hybrid organic-metallic charring agent (OTCA@ZnO) had been prepared. The flame retardancy and thermal stability of EVA composites containing APP and OTCA@ZnO were studied through LOI, UL-94, cone test and TGA. The residues were studied to explore the flame-retardant mechanism by SEM, FTIR, LRS and XPS. APP/HOTCA/ZnO system as a comparing subject of APP/OTCA@ZnO system was also systematically investigated to identify the more efficient flame-retardant action of the hybrid system.

There are many special features of this work. Firstly, the flame retardant interacts with the metal oxide and exists in the same molecular unit, which realizes the organic unification of modification

and catalysis, and greatly improves the flame retardant efficiency. Secondly, the surface modification of the metal oxide is used to solve the problem of compatibility of inorganic metals. The problem that the organic small molecule flame retardant is easy to migrate is also solved. Thirdly, the dispersibility of the nano metal oxide is improved, and the structure of the organic char-forming agent encapsulating the nanoparticle effectively improves the mechanical properties of the polymer. At the same time, cyanuric chloride is a kind of C and N alternating benzene ring-like structure, which has good char forming ability, and silicon-containing compounds are types of environment-friendly and highly efficient flame retardants which could form a protective carbonaceous layer with excellent barrier effect at high temperature.

Comment 4: Clarify SCTCA.

Response: Thanks for the referee's kind advice. OTCA meant the organic triazine charring agent. The structure of OTCA was added into the revised manuscript as following scheme:

Scheme 1. Route for the synthesis of OTCA

Comment 5: How to determine the feeding ratio for the preparation of SCTCA?

Response: Thanks for the reviewer's question. Based on the different reactivity of the three chlorine atoms in the on the triazine ring, the first one was substituted by γ -aminopropyltriethoxy silane at 0 °C, and the remaining two chlorine atoms are replaced by an amino group on two ethylene diamines at 50 °C and 100 °C, respectively. So the molar ratio of γ -aminopropyltriethoxy, CYC and EDA is 1:1:1.2.

Comment 6: Describe the reaction mechanism for the preparation of SCTCA with a schematic.

Response: Thanks for the referee's suggestion. Based on the different reactivity of the three chlorine atoms in the on the triazine ring, the first one was substituted by γ -aminopropyltriethoxy silane at 0 °C, and the remaining two chlorine atoms are replaced by an amino group on two ethylene diamines at 50 °C and 100 °C, respectively. The reaction mechanism of OTCA was added into the revised manuscript and was as following:

Scheme 1. Route for the synthesis of OTCA

Comment 7:2.2, what was used to adjust pH value?

Response: Thanks for the reviewer's question. 5 wt% ammonia solution was used to adjust pH value.

Comment 8:P4, "Scheme 1 presented the routes of synthesis for SCTCA and SCTCA@ZnO", but the route for SCTCA synthesis was missed.

Response: Thanks for the referee's kind suggestion. In order to more clearly represent the structure of the compound, we have added the scheme 1 as following:

Scheme 1. Route for the synthesis of OTCA

Scheme 2. Route for the synthesis of OTCA@ZnO

Comment 9: The full names for all abbreviations should be given when they appear for the first time, such as TGA, LRS and PHRR.

Response: Thanks for the referee's kind advice. We have carefully checked the manuscript and the full names for all abbreviations had been given when they appear for the first time, such

as TGA, LRS and PHRR, marked in green in the revised manuscript.

Comment 10:P7, what are HSCTCFA, HSCTFA/ZnO and SCTCFA-ZnO? They are not consistent with the expressions in the text and table! Please use the same expressions in full text, figures and tables!

Response: Thank you very much for your advice. For the abbreviation of the referred compounds in the manuscript, it was our mistake, leading to unwanted confusing. In the modified manuscript, all the compounds have given the modified abbreviations marked in green, which were consistent in full text, figures and tables. The modified abbreviations were as follows:

OTCA: organic triazine charring agent.

HOTCA: hydrolyzed OTCA.

OTCA@ZnO: organic triazine charring agent hybrid with nano-ZnO.

HOTCA/ZnO: physical blending of HOTCA and nano-ZnO.

Comment 11:Page 3 line 38-42 “Firstly, 20 mL DMAc solution containing 32 g γ -aminopropyltriethoxy silane was added into athree-necked flask equipped with a mechanical stirrer. Then, 27 g

CYC and 14.6 g TEA in 20 mLDMAC was slowly added dropwise with vigorously stirring in an external ice bath”. Was it ice bath or ice-water bath? What was the temperature? In addition, the reaction temperature was gone up to 50 °C after three hours. If ice bath was used, how to increase the temperature? The description is too confusing.

Response: Thank you very much for your question. It was our fault that the description of the reaction condition was vague. “an external ice bath” indicated ice-water bath and the temperature was kept at 0 °C for three hours. Then the reaction temperature was gone up to 50 °C in oil bath. According to your valuable suggestions, this section has been added in the revised manuscript marked in green.

Comment 12:Page 3 line 51-52 “After dried to a constant weight under vacuum, the intermediate SCTCA (light yellow powder) was obtained”. What was the drying temperature?

Response: Thank you very much for your question. It was also our fault that the drying temperature of the products was not given in the manuscript. The drying temperature was 100 °C. This has been modified in the revised manuscript and marked in green.

Comment 13: Page 4 Line 3-4, what is the purpose for activation of nano ZnO? Is there any reference?

Response: Thank you very much for your question, which is very valuable. The purpose for activation of nano-ZnO was to produce hydroxyl groups on its surface under high temperature conditions, so that the hydroxyl groups on the surface of the nano-ZnO could be condensed with the hydroxyl groups on OTCA (SCTCA). There were lots of references about the surface modification of nanometer oxide particle and in fact, the activation to generate hydroxyl groups on the surface of the nanometer oxide particles, such as nano-ZnO, nano-TiO₂, nano-SiO₂ and so on, was a common and typical method in the similar studies. For example, Chen-Chi M. Ma et al [1] has mentioned "Surface modified with coupling agents: Nano scale zinc oxide powders were dried for 24 h, and then put into the reactor. VTES (C₈H₁₈O₃Si) and PTES (C₁₂H₂₀O₃Si) were used as the coupling agents and mixed with nanoscale zinc oxide powder, respectively. The weight ratio of the coupling agent and nanoscale zinc oxide powder was 20:1. THF was used as solvent and small amount acetic acid potassium was used as catalyst, and the mixture was stirred

at 60°C for 24 h. After the reaction was complete, the mixture was separated by centrifuging three times, and then dried in a vacuum oven.”

[1] Ma CCM, Chen YJ, Kuan HC. 2005 Polystyrene Nanocomposite Materials—Preparation, Mechanical, Electrical and Thermal Properties, and Morphology. *J.Appl.Pl.Polym.Sci.* **100**, 508-515. (doi:10.1002/app.23221)

Comment 14:Page 4 line 10, “The solid was washed three times with hot water”, why was hot water used?

Response: Thank you very much for your comment. When the crude products after reaction were post-processed, we considered the feedstock may have the greater solubility in hot water than cold water according to the common sense, and thus hot water was chose to wash the crude product. Maybe cold water was also usable, which has not been tried in this experiment.

Comment 15: The authors should characterize the dispersion of the fillers in the composites by SEM or TEM.

Response: Thank you very much for your kindly suggestion. According to your suggestion, the dispersion of the fillers in the

composites has been characterized using SEM and the results and discussion have been added into the modified manuscript marked in green. Here is an excerpt.

3.5. Dispersion state and compatibility of fillers

The dispersion state and compatibility of APP/OTCA@ZnO (Figure 10) and APP/HOTCA/ZnO (Figure 10) in EVA was investigated by SEM. Comparing Figure 10 (a) and (c), it could be seen that EVA/IFR-6 had plenty of small holes and bulges, apparently as a result of poor compatibility of pristine nano-ZnO particles with the matrix, while EVA containing OTCA@ZnO didn't have this appearance. The dispersion around APP was presented in Figure 10 (b) and d and it could be observed that the compatibility between the char forming agent and APP was not bad. From Figure 10 (d), it was worth noting that a large amount of nano-zinc oxide was accumulated around APP and the charring agent, and most of them existed in the form of agglomerate. Such a distribution not only deteriorated the compatibility of the fillers with the matrix, but also many nano-ZnO were not sufficiently in contact with APP and the char-forming agent, which resulted in a significant reduction of flame-retardant efficiency.

In contrast, the hybrid nano-zinc oxide was encapsulated by OTCA, showing a better compatibility in the matrix and APP. In this way, nano-zinc oxide together with the char forming agent could be more uniformly dispersed in matrix, thereby improving the flame retardant efficiency. This distribution also explained why the initial decomposition temperature of EVA/IFR-3 was higher than that of EVA/IFR-6.

Figure 10. SEM images of the freeze-fractured surface EVA/IFR-3 (a and b) and EVA/IFR-6 (c and d).

Thanks very much for your kind work and consideration of our work again.

We greatly appreciate the efficient, professional and rapid processing of our manuscript by your team. If there is anything else we should do, please do not hesitate to let us know.

Thanks again and best regards.

Yours sincerely,

Prof. Bo Xu

E-mail: xubo@btbu.edu.cn